# Long-term changes as oil palm plantation age simplify the structure of host-parasitoid food webs

**Akhmad Rizali**[1]*, **Sri Karindah**[1], **Nur Ainy**[1], **Muhamad Luthfie Tri Meiadi**[1], **Muhammad Iqbal Tawakkal**[2], **Bambang Tri Rahardjo**[1], **Damayanti Buchori**[2]

**1** Faculty of Agriculture, Department of Plant Pests and Diseases, University of Brawijaya, Malang, East Java, Indonesia, **2** Faculty of Agriculture, Department of Plant Protection, IPB University, Bogor, West Java, Indonesia

* arizali@ub.ac.id

**Data Availability Statement:** All relevant data are within the paper and its Supporting Information files.

## Abstract

Understanding host-parasitoid food webs, as well as the factors affecting species interactions, is important for developing pest management strategies in an agroecosystem. This research aimed to study how the long-term change in oil palm plantations, specifically the tree age, affect the structure of host-parasitoid food webs. The field research was conducted on an oil palm plantation located in Central Kalimantan and Jambi Province, Indonesia. In Central Kalimantan, we conducted observations of lepidopteran larvae and parasitoid wasps at different tree ages, ranging from 3 to 18 years old. For tree ages from 3 to 10 years, observations of host-parasitoid food webs were conducted by collecting the lepidopteran larvae using a hand-collection method in each oil palm tree within a hundred trees and they were later reared in the laboratory for observing the emerging parasitoids. The fogging method was applied for trees aged 12 to 18 years because the tree height was too high, and hand-collection was difficult to perform. To compare host-parasitoid food webs between different regions, we also conducted a hand-collection method in Jambi, but only for trees aged 3 years old. The food-web structure that was analyzed included the species number of lepidopteran larvae and parasitoid wasps, linkage density, and interaction diversity. We found 32 species of lepidopteran pests and 16 species of associated parasitoids in Central Kalimantan and 12 species of lepidopteran pests, and 11 species of parasitoids in Jambi. Based on the GLM analysis, tree age had a negative relationship with the species number of lepidopteran larvae and parasitoids as well as linkage density and interaction diversity. Different geographical regions showed different host-parasitoid food web structures, especially the species number of lepidopteran larvae and interaction diversity, which were higher in Central Kalimantan than in Jambi. However, some parasitoids can be found across different tree ages. For example, *Fornicia* sp (Hymenoptera: Braconidae) was recorded in all ages of oil palm sampled. Results of the GLM analysis showed that the abundance of *Fornicia* sp and its host (lepidopteran larvae abundance) were not affected by the tree age of the oil palm. In conclusion, the long-term change in oil palm plantations simplifies the structure of host-parasitoid food webs. This highlights the importance of long-term studies across

**Funding:** AR received funding from Badan Pengelola Dana Perkebunan Kelapa Sawit (Indonesian Oil Palm Estate Fund Agency), Ministry of Finance of the Republic of Indonesia, award number: Year 2016 and Ministry of Research and Technology, Republic of Indonesia through the grant program of World Class Research, year 2021, award number: 108/SP2H/LT/DRPM/2021. The funders had no role in study design, data collection and analysis, decision to publish, or preparation of the manuscript.

**Competing interests:** The authors have declared that no competing interests exist.

geographical regions for a better understanding of the consequences that wide monoculture oil palm plantations have on biological control services.

## Introduction

Understanding the ecological consequences of species interactions plays a pivotal role in identifying the factors that affect the decrease in ecosystem services in an agroecosystem. For instance, the outbreak of phytophagous insects can be driven by changes in trophic interactions between plants and herbivores or between prey and predators [1], as a consequence of agricultural practices, especially pesticide application [2]. However, agricultural practices can also interact with habitat conditions around agricultural areas, resulting in different interactions between herbivores and natural enemies [3]. Research by [4] revealed that agricultural landscape management through the enhancement of insect spillover between crop and non-crop areas may involve desirable and undesirable interactions. Therefore, the management of species interactions in the agroecosystem involves not only understanding the simple food chains of herbivores and natural enemies [5] but also considering other factors, such as community structure and ecosystem function, to achieve the stability of interaction networks as an indicator of sustainable agriculture [6].

In tropical ecosystems, changes in species interactions are influenced by habitat modification [7]. This is due to all species being linked in networks of mutualistic and antagonistic interactions [8]. The importance of conserving these interactions and associated processes, as well as the component species [9] associated with species interactions, such as pollination and biological control, supports ecosystem services [10]. Research on changing patterns of interactions in ecological communities has been hampered by the lack of proper analytical tools and by a failure to detect the presence of interspecific interactions [11]. Previous food-web studies considered all interactions as equal and qualitative properties used to describe food webs were found to be sensitive to variations in sampling effort [12,13].

This paper investigates how the long-term change of oil palm plantations, specifically the tree age, affects host-parasitoid food webs. We used the quantification method of interactions at the community scale [14], which provides a more robust quantification of community structure and insights into the dynamic processes that structure ecological communities [15]. This approach is most commonly applied to networks of trophic interactions such as quantitative food webs [14,16]. The information contained in food webs can be summarized in qualitative and quantitative metrics using bipartite graphs, including the number of herbivore and parasitoid species, linkage density (qualitative counterparts), and interaction diversity (quantitative measures of diversity) [7,17].

As a consequence of the monoculture system, oil palm plantations experience attacks by insect pests [18] that can cause significant yield losses [19,20]. However, current management practices have negative impacts on biodiversity, which might disturb the natural regulation of pests and diseases [21], such as pesticide applications that have non-target effects on natural enemies and cause a simplification in the interaction between pests and their natural enemies [22,23]. Unfortunately, information about the effect of the long-term change of oil palm plantations on host-parasitoid food webs is lacking. This is concerning because the species composition and species richness of communities in transformed ecosystems may not remain stable even if management does not change [24]. Increasing the age of oil palms tends to increase the species richness of canopy arthropods [25], flower-visiting insects [26], and ground-foraging ant communities [27].

This research aimed to study the structure of host-parasitoid food webs in oil palm plantations and investigate the effect of long-term changes in oil palm plantations on the trophic interaction between herbivores and parasitoids. The research was conducted in oil palm plantations in Central Kalimantan and Jambi Province, Indonesia. We observed herbivores and parasitoids in different tree ages of oil palm, ranging from 3 to 18 years old. We tested the hypotheses that the tree age of oil palm and different geographical regions affect the structure of host-parasitoid food webs in oil palm plantations. Different geographical regions are expected to shape different host-parasitoid food webs, even in similar latitudes and climate conditions.

## Materials and methods

### Research location

The research was conducted in oil palm plantations located in West Kotawaringin Regency, Central Kalimantan (111˚49' E, 2˚18' S) and Batang Hari Regency, Jambi (103˚17' E, 1˚50' S), Sumatra, Indonesia. To avoid the effect of latitudinal gradients [28], we selected Central Kalimantan and Jambi because these two regions are located at similar latitudes (around 2˚ S) and have similar topography, primarily lowland areas in agroclimatic zone II (rainfall 1750–3000 mm, 1–2 dry months, and sunshine duration 6 h/day) [23]. In Central Kalimantan, we selected 9 plots with different tree ages of oil palms, i.e., 3, 6, and 10 years, to observe host-parasitoid food webs, and 10 plots with tree ages of oil palms ranging from 12 to 18 years for the observation of parasitoid wasp diversity and lepidopteran larvae abundance.

To compare with other geographic areas, we also conducted observations of host-parasitoid food webs in an oil palm plantation in Jambi, Sumatra. In this area, we selected three plots with tree age 3 years and recorded host-parasitoid food webs, using the same methods applied in Central Kalimantan.

### Observation of host-parasitoid food webs

To observe the host-parasitoid food webs, a hundred oil palm trees (10 trees x 10 trees) were selected per plot as a sampling unit. Lepidopteran larvae were collected using a hand-collection method in each oil palm tree for all hundred trees. We searched for lepidopteran larvae for a total of 15 minutes per tree, spending 5 minutes in the top, middle, and bottom sections of each canopy. All instars of lepidopteran larvae were collected and placed individually into a plastic container containing oil palm leaves, which were then brought to the laboratory for rearing and observation of the emerging parasitoid. In each plot, the observation and collection were conducted monthly for three months. In Central Kalimantan, lepidopteran pests were sampled from March to May 2017, while in Jambi, the observations were conducted from April to June 2017.

In the laboratory, lepidopteran larvae were reared in plastic containers and placed on a shelf at room temperature, which was equipped with lubricating oil to protect them from ants. Lepidopteran larvae were observed daily to record the stage of development, disease status, and parasitoid or moth emergence. The parasitized larvae were separated and intensively observed until the adult parasitoid emerged.

### Observation of parasitoid wasp diversity and lepidopteran larvae abundance

We selected oil palm trees above 10 years old to observe parasitoid wasp diversity and lepidopteran larvae abundance using the fogging method. This choice was made because, at these

ages, the oil palm trees are too high and difficult to observe using the hand-collection method. In addition to the abundance of all parasitoid wasps, we also selected and calculated the abundance of *Fornicia* sp. as an indicator of the most common species of parasitoids, since this species was recorded across all tree ages of oil palm.

In each oil palm field, plots with a size of 50 m x 70 m were established, and six oil palm trees with similar heights and canopy sizes were randomly selected within these plots as sampling units. The fogging method used 2.5% pyrethroid insecticides (lambda-cyhalothrin), which were applied using a PulsFOG K-22 Bio fogging machine. Fogging was performed at 06:00 AM during the daytime when the wind speed was the lowest. At each sample unit, fogging was performed for approximately 10 minutes or until the whole canopy was entirely covered with insecticide fog. Approximately 60 minutes after fogging or until the insecticide fog disappeared, the killed insects were collected from an 8 m x 8 m sheet of white canvas that had been placed under each tree. As the sheet was placed next to the tree, collections would likely include insects collected from both the oil palm canopy and epiphytes growing on the trunk. For each plot, fogging was repeated three times in different months from February to April 2017.

## Observation of understorey vegetation

To provide information about habitat conditions on each plot (both for hand-collection and fogging methods), understorey vegetation diversity was observed using the visual observation method. We determined 10 random points in each plot, each with a size of 1 m x 1 m. At each point, we calculated the number of understorey species and the cover of vegetation per quadrat. The cover of vegetation was measured based on the percentage cover of all understorey species in 1 m x 1 m. Furthermore, each vegetation species was sampled and photographed for later identification in the laboratory. The identification of vegetation was conducted using available identification books [e.g. 29].

## Insect identification

The insect specimens, both from the hand-collection method and fogging method, were initially sorted and identified to the order level. Lepidopteran larvae were identified using an available reference book [30], while parasitoid wasps were identified to the family level with the aid of entomological handbooks [31–33]. They were further separated into morphospecies levels based on morphological characteristics. If possible, some morphospecies were identified at the genus level using available references [e.g. 34,35]. From the fogging data, we calculated the abundance of lepidopteran larvae, all parasitoid wasps, and *Fornicia* sp per plot.

## Food webs construction and data analysis

To assess the effect of tree age and host abundance on host-parasitoid food webs, we used quantitative metrics with well-known qualitative counterparts (linkage density) and quantitative measures of interaction diversity (Shannon diversity) [7,17]. Linkage density is computed as the total number of trophic links and the number of taxa in the food web [36], while Shannon diversity is calculated based on interaction diversity (i.e. network entries) [17]. Based on data from the hand-collection method, we constructed trophic interactions between lepidopteran pests (lower bars) and their parasitoids (upper bars). Data from all 100 trees in each plot for all time periods were pooled prior to analyses (see S1 Table for Central Kalimantan and S2 Table for Jambi). Trophic interaction networks of lepidopteran pests and their parasitoids for each tree age of oil palm and different regions were quantified by using the bipartite ecological network [17]. From the analysis results, we selected the metrics of trophic interaction

networks, including linkage density, Shannon diversity, number.of.species.HL (species number of parasitoids in the network) and number.of.species.LL (species number of lepidopteran larvae in the network).

Assuming that each plot was independent, we used a generalized linear model (GLM) without interactions [37] to study the effect of long-term changes in oil palm plantations on host-parasitoid food webs and used a quasipoisson distribution to account for overdispersion. Long-term changes in oil palm plantations were indicated by the tree age of oil palm, and the number of species and cover of understorey vegetation as explanatory variables. From the data of the hand-collection methods, we studied the effect of explanatory variables on the number. of.species.HL number.of.species.LL, linkage density, and Shannon diversity. While from the data of the fogging method, we studied the effect of explanatory variables on all parasitoid wasps, *Fornicia* sp and lepidopteran abundance.

To study the effect of different regions on host-parasitoid food webs, we used analysis of variance (ANOVA) to compare number.of.species.HL number.of.species.LL, linkage density, and Shannon diversity in the same tree age of oil palm (3 years old) between Central Kalimantan and Jambi. All analyses were performed using the R statistic [38].

## Results

### Diversity and food-web structure of lepidopteran pests and parasitoids in oil palm plantation

Based on the hand-collection method, we found 32 species and 5,522 individuals of lepidopteran pests and 16 species and 212 individuals of parasitoid wasps from different tree ages in oil palm plantations in Central Kalimantan (Table 1). Out of all collected lepidopteran larvae, only 100 larvae (0.02%) were parasitized by parasitoids with the lowest percent parasitization found in Family Psychidae (1.7%) (Table 2). The results of bipartite analysis showed that only 10 species of lepidopteran pests (LL) are parasitized by 16 species of parasitoids (HL). The highest linkage density (2.072) and Shannon diversity were found in plots with oil palm trees aged 3 years. The highest abundance of parasitoid wasps was *Fornicia* sp (Hymenoptera: Braconidae), which attacked *Darna trima* (65 individuals) with a parasitization level of 20% and *Darna diducta* (15 individuals) with a parasitization level of 11.54% (Table 2).

**Table 1. Plot characteristics, species richness (S) and abundance (N) of lepidopteran larvae and parasitoids, as well as metrics of food webs based on hand-collecting methods in oil palm plantation in Central Kalimantan.** Plot characteristics include the tree age of oil palm and the diversity and cover of understorey vegetation.

| No | Plot | Tree age (year) | Vegetation | | Lepidopteran larvae | | Parasitoid | | Food webs metrics | | | |
|---|---|---|---|---|---|---|---|---|---|---|---|---|
| | | | Species | Cover (%) | S | N | S | N | LL | HL | LD | H' |
| 1. | P31 | 3 | 17 | 52.6 | 22 | 1,611 | 10 | 40 | 5 | 10 | 2.006 | 2.233 |
| 2. | P32 | 3 | 15 | 56.0 | 23 | 1,780 | 6 | 26 | 5 | 6 | 2.072 | 2.012 |
| 3. | P33 | 3 | 9 | 28.9 | 9 | 346 | 5 | 17 | 5 | 5 | 1.196 | 1.316 |
| 4. | P61 | 6 | 23 | 37.1 | 12 | 563 | 4 | 22 | 5 | 4 | 1.655 | 1.463 |
| 5. | P62 | 6 | 21 | 34.1 | 13 | 576 | 6 | 21 | 6 | 6 | 1.264 | 1.766 |
| 6. | P63 | 6 | 24 | 55.4 | 9 | 242 | 3 | 11 | 2 | 3 | 1.175 | 0.600 |
| 7. | P101 | 10 | 21 | 43.9 | 13 | 76 | 1 | 5 | 1 | 1 | 1.000 | 0.000 |
| 8. | P102 | 10 | 16 | 72.4 | 6 | 133 | 1 | 3 | 1 | 1 | 1.000 | 0.000 |
| 9. | P103 | 10 | 13 | 12.2 | 8 | 195 | 3 | 67 | 4 | 3 | 1.015 | 0.418 |
| | | | | Total | 32 | 5,522 | 16 | 212 | 10 | 16 | | |

Metrics of host-parasitoid food webs include LL: Species number of lepidopteran larvae in the network, HL: Species number of parasitoids in the network, LD: Linkage density, H': shannon diversity.

**Table 2. Abundance of lepidopteran larvae and their parasitoids from nine plots of oil palm plantations in Central Kalimantan.**

| Host (Family/Species) | No. larvae | No. (%) parasitized | Parasitoid | No. parasitoid | Type |
|---|---|---|---|---|---|
| Lymantriidae | 127 | 6 (4.7) | | | |
| Lymantriidae sp4 | 12 | 2 (16.7) | Encyrtidae sp3 | 1 | Solitary |
| | | | Eulophidae sp9 | 1 | Solitary |
| Lymantriidae sp8 | 30 | 2 (6.7) | Braconidae sp32 | 2 | Solitary |
| Lymantriidae sp11 | 8 | 1 (12.5) | Ichneumonidae sp45 | 1 | Solitary |
| Lymantriidae sp12 | 9 | 1 (11.1) | Eulophidae sp8 | 5 | Gregarious |
| Limacodidae | 87 | 7 (8.1) | | | |
| *Darna diducta* | 26 | 5 (19.2) | Bethylidae sp1 | 1 | Solitary |
| | | | *Fornicia* sp | 65 | Gregarious |
| | | | Eurytomidae sp1 | 5 | Gregarious |
| *Darna trima* | 10 | 2 (20.0) | *Fornicia* sp | 15 | Gregarious |
| Psychidae | 5263 | 87 (1.7) | | | |
| *Clania tertia* | 981 | 11 (1.1) | Braconidae sp5 | 1 | Solitary |
| | | | Braconidae sp137 | 1 | Solitary |
| | | | *Elasmus* sp | 8 | Gregarious |
| | | | Eulophinae sp2 | 1 | Solitary |
| | | | Ichneumonidae sp8 | 2 | Solitary |
| | | | Ichneumonidae sp30 | 1 | Solitary |
| | | | Ichneumonidae sp45 | 1 | Solitary |
| *Mahasena corbetti* | 1,312 | 28 (2.1) | *Aulosaphes* sp | 40 | Gregarious |
| | | | Braconidae sp137 | 1 | Solitary |
| | | | *Elasmus* sp | 6 | Gregarious |
| | | | Ichneumonidae sp45 | 2 | Solitary |
| *Metisa plana* | 2,568 | 36 (1.4) | *Aulosaphes* sp | 8 | Gregarious |
| | | | Braconidae sp137 | 3 | Solitary |
| | | | *Elasmus* sp | 14 | Gregarious |
| | | | Eulophinae sp2 | 4 | Solitary |
| | | | Ichneumonidae sp8 | 1 | Solitary |
| | | | Ichneumonidae sp30 | 2 | Solitary |
| | | | Ichneumonidae sp45 | 3 | Solitary |
| *Pteroma pendula* | 402 | 12 (3.0) | Braconidae sp5 | 1 | Solitary |
| | | | *Aulosaphes* sp | 1 | Solitary |
| | | | Braconidae sp137 | 4 | Solitary |
| | | | Eulophinae sp2 | 1 | Solitary |
| | | | Eulophidae sp10 | 10 | Gregarious |

% parasitized is the percentage of parasitized larvae from the total number of collected larvae.

The GLM showed that only the tree age of oil palm was a significant predictor ($P < 0.05$) of the structure of host-parasitoid food webs in oil palm plantations (Table 3). A negative estimate value indicated that increasing the tree age of oil palm caused a decrease in the species number of lepidopteran larvae and parasitoids, resulting in a simpler food web structure (Fig 1). Although increasing the tree age of oil palm led to a decrease in the species number of parasitoids, several parasitoids were consistently recorded across all tree ages of oil palm such as *Fornicia* sp, *Aulosaphes* sp (Hymenoptera: Braconidae), and *Elasmus* sp (Hymenoptera: Eulophidae) (Figs 1 and 2). Meanwhile, lepidopteran pests were found across all tree ages of oil palm and included *Clania tertia*, *Darna diducta*, *Mahasena corbetti*, and *Metisa plana* (Fig 1).

**Table 3. Generalized linear models relating the abundance of lepidopteran larvae and the structure of host-parasitoid food webs to the tree age of oil palm, diversity, and cover of understorey vegetation as predictors from nine plots of oil palm plantation in Central Kalimantan.**

| Variable | Estimate | SE | P |
|---|---|---|---|
| Lepidopteran larvae | | | |
| (Intercept) | 6.939 | 1.01 | 0.001 |
| Tree age | -0.361 | 0.13 | 0.035* |
| Vegetation diversity | 0.036 | 0.06 | 0.546 |
| Vegetation cover | 0.015 | 0.02 | 0.434 |
| Lepidopteran larvae (LL) | | | |
| (Intercept) | 2.853 | 0.76 | 0.000 |
| Tree age | -0.176 | 0.08 | 0.037* |
| Vegetation diversity | 0.033 | 0.04 | 0.438 |
| Vegetation cover | -0.026 | 0.02 | 0.090 |
| Parasitoids (HL) | | | |
| (Intercept) | 2.716 | 0.62 | 0.007 |
| Tree age | -0.239 | 0.07 | 0.023* |
| Vegetation diversity | 0.034 | 0.03 | 0.378 |
| Vegetation cover | -0.012 | 0.01 | 0.367 |
| Linkage density (LD) | | | |
| (Intercept) | 0.515 | 0.32 | 0.172 |
| Tree age | -0.085 | 0.03 | 0.027* |
| Vegetation diversity | 0.011 | 0.02 | 0.533 |
| Vegetation cover | 0.003 | 0.00 | 0.596 |
| Shannon diversity (H') | | | |
| (Intercept) | 2.394 | 0.69 | 0.018 |
| Tree age | -0.356 | 0.10 | 0.015* |
| Vegetation diversity | 0.064 | 0.04 | 0.172 |
| Vegetation cover | -0.012 | 0.01 | 0.418 |

Metrics of host-parasitoid food webs include LL: Species number of lepidopteran larvae in the network, HL: Species number of parasitoids in the network, LD: Linkage density, H': Shannon diversity.
* $P < 0.05$.

In addition, we found 12 species of lepidopteran pests and 11 species of parasitoids in Jambi for 3 years old oil palm trees. Only 7 species of lepidopteran pests are parasitized by parasitoids (Table 4), with the lowest percent parasitization found in the Family Lymantriidae (15.6%). Compared to the same age in Central Kalimantan, the species number of lepidopteran larvae and Shannon diversity are significantly higher in Central Kalimantan than in Jambi, while the species number of parasitoids and linkage density are not different (Table 5).

## Effect of host abundance, tree age, and understorey vegetation on parasitoid abundance

Based on the fogging method, we found 229 species and 2195 individuals of parasitoid wasps, with the most dominant parasitoids belonging to the Family Braconidae (131 species and 1229 individuals) (see S3 Table). A braconid species, *Fornicia* sp, commonly recorded from the hand-collection method, was also recorded in the fogging method across all tree ages of oil palm ranging from 12 to 18 years (Table 6). The results of the GLM analysis showed that the abundance of lepidopteran larvae, all parasitoid wasps and *Fornicia* sp were not affected by tree age or the diversity and cover of understorey vegetation ($P > 0.05$) (Table 7).

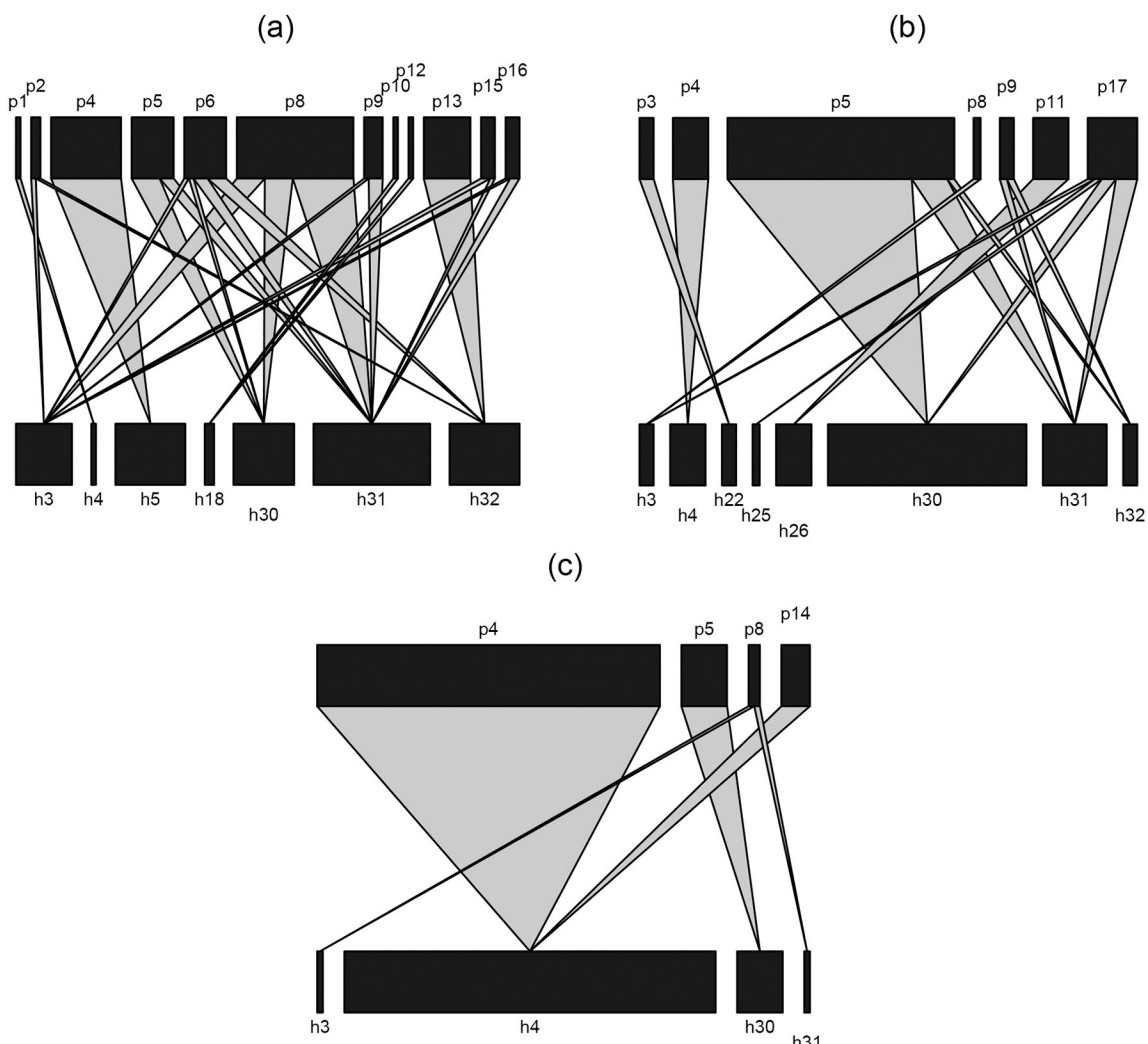

**Parasitoid (top bar)**: p1: Bethylidae sp1, p2: Braconidae sp5, p3: Braconidae sp32, p4: *Fornicia* sp, p5: *Aulosaphes* sp, p6: Braconidae sp137, p8: *Elasmus* sp, p9: Elasmidae sp2, p10: Encyrtidae sp3, p11: Eulophidae sp8, p12: Eulophidae sp9, p13: Eulophidae sp10, p14: Eurytomidae sp1, p15: Ichneumonidae sp8, p16: Ichneumonidae sp30, p17: Ichneumonidae sp45

**Lepidopteran pest (bottom bar)**: h1: *Birthamula chara*, h2: *Birthosea bisura*, h3: *Clania tertia*, h4: *Darna diducta*, h5: *Darna trima*, h6: Geometridae sp1, h7: Geometridae sp2, h8: Geometridae sp3, h9: Geometridae sp4, h10: Geometridae sp5, h11: Geometridae sp6, h12: Geometridae sp7, h13: Geometridae sp8, h15: Lymantriidae sp1, h16: Lymantriidae sp2, h17: Lymantriidae sp3, h18: Lymantriidae sp4, h19: Lymantriidae sp5, h20: Lymantriidae sp6, h21: Lymantriidae sp7, h22: Lymantriidae sp8, h23: Lymantriidae sp9, h24: Lymantriidae sp10, h25: Lymantriidae sp11, h26: Lymantriidae sp12, h27: Lymantriidae sp13, h28: Lymantriidae sp14, h29: Lymantriidae sp15, h30: *Mahasena corbetti*, h31: *Metisa plana*, h32: *Pteroma pendula*, h33: *Setora nitens*

**Fig 1.** Host-parasitoid food webs in different tree ages of oil palm: (a) 3 years (n = 3 plots), (b) 6 years (n = 3 plots), and (c) 10 years (n = 3 plots). For each web, the lower bars (LL) represent host (lepidopteran pests) abundance, and the upper bars (HL) represent parasitoid abundance, drawn at different scales. Linkage width indicates the frequency of each trophic interaction. Some species of lepidopteran pests that were not parasitized by parasitoid wasps are not shown in the bars.

(a)

(b)

(c)

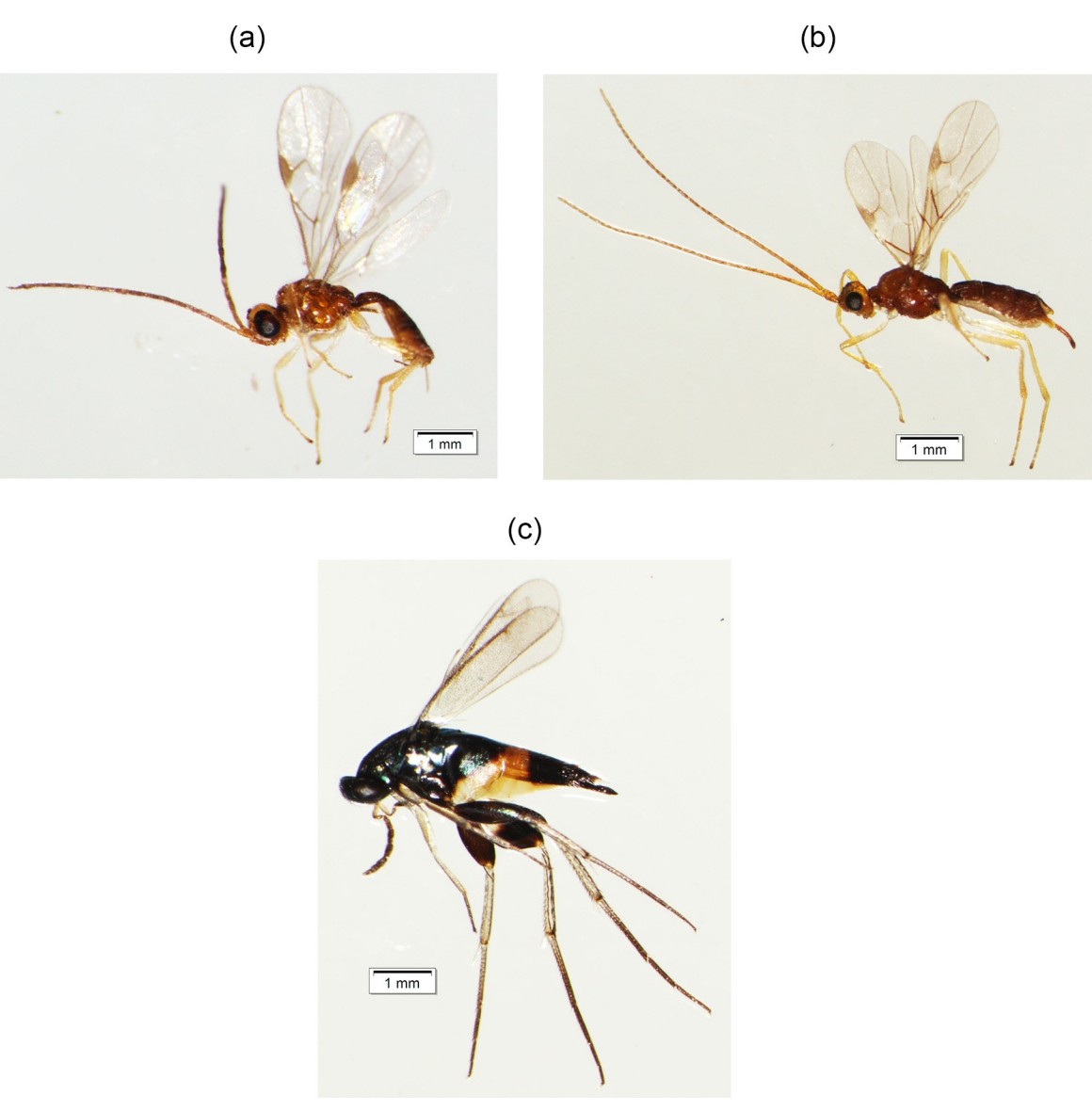

**Fig 2. Species of parasitoid wasps that were always found in different tree ages of oil palm in Central Kalimantan.** (a) *Fornicia* sp, (b) *Aulosaphes* sp, and (c) *Elasmus* sp.

## Discussion

The research results showed that increasing the tree age of oil palm simplifies the structure of host-parasitoid food webs in oil palm plantations. Both linkage density and Shannon diversity were affected by the tree age of oil palm, but no significant differences in linkage density were detected between geographical regions. This is due to linkage density being strongly affected by species richness [12], and the absence of significant differences in richness between regions might have led to the consistency of this metric. In contrast, interaction diversity based on the Shannon diversity has a significant difference between tree age and geographical regions. The result is similar to [7], which suggests that interaction diversity, as a quantitative metric, can detect differences in food webs better than linkage density, which is a qualitative metric. The decreasing Shannon diversity of the food web indicates that the uniformity of host and

**Table 4. Abundance of lepidopteran pests and their parasitoids from three plots in oil palm plantations in Jambi.**

| Host | No. larvae | No. (%) parasitized | Parasitoid | No. parasitoid | Type |
|---|---|---|---|---|---|
| Geometridae | 6 | 2 (33.3) | | | |
| Geometridae sp1 | 6 | 2 (33.3) | Eulophidae sp3 | 10 | Gregarious |
| | | | Ichneumonidae sp6 | 1 | Solitary |
| Limacodidae | 48 | 12 (25.0) | | | |
| *Darna trima* | 8 | 1 (12.5) | Ichneumonidae sp5 | 1 | Solitary |
| *Parasa lepida* | 5 | 4 (80.0) | Braconidae sp2 | 59 | Gregarious |
| *Setora nitens* | 33 | 7 (21.2) | Braconidae sp1 | 5 | Gregarious |
| | | | Eulophidae sp2 | 43 | Gregarious |
| | | | Eurytomidae sp1 | 1 | Solitary |
| Lymantriidae | 32 | 5 (15.6) | | | |
| Lymantriidae sp1 | 23 | 2 (8.7) | Ichneumonidae sp1 | 1 | Solitary |
| | | | Ichneumonidae sp2 | 1 | Solitary |
| Lymantriidae sp2 | 1 | 1 (100) | Braconidae sp3 | 39 | Gregarious |
| Lymantriidae sp3 | 8 | 2 (25.0) | Eulophidae sp1 | 224 | Gregarious |

% parasitized is the percentage of parasitized larvae out of the total number of collected larvae.

parasitoid communities tends to become simpler with the increasing tree age of oil palm. This might be related to the increasing height of oil palm trees, a result of tree growth that affects the microclimate and understorey vegetation [23] and inhibits certain insects from accessing the canopy [39]. However, the parasitism can be higher in the old age of oil palm as a consequence of a lower interaction network [7]. This means that this age has the strongest top-down control, where food webs are dominated by a single link [40].

The results also showed that increasing the tree age of oil palm (above 12 years old) did not affect the abundance of either lepidopteran larvae or the specialist parasitoid, *Fornicia* sp. The different results could be due to different methods and metrics used, with direct collection and raising of parasitoids likely to be more sensitive to the effects of oil palm age. It might also be due to older oil palm plantations tending to have similar environmental conditions once the canopy has closed, compared to differences in the environment between years in younger oil palm plantations. The results rejected our hypothesis that increasing the tree age of oil palm changed the plant architecture and shaped different habitat conditions that would facilitate parasitoid wasp communities in oil palm plantations. Although the development of plant architecture can facilitate other biodiversity, such as ants, for nesting site availability and microhabitat structure [41,42], the tree growth of oil palm does not support arthropod communities [e.g. 39], including lepidopteran larvae and specialist parasitoid wasps.

**Table 5. The different structures of host-parasitoid food webs collected from oil palm plantations between Central Kalimantan (n = 3 plots) and Jambi (n = 3 plots).**

| Parameter | Central Kalimantan | Jambi | Significant |
|---|---|---|---|
| Lepidopteran larvae (LL) | 5.0±0.00 | 2.7±1.15 | $F_{1,4} = 12.25, P = 0.025$* |
| Parasitoids (HL) | 7.0±2.65 | 4.0±2.65 | $F_{1,4} = 1.929, P = 0.237$ |
| Linkage density (LD) | 1.758±0.49 | 1.066±0.09 | $F_{1,4} = 5.857, P = 0.073$ |
| Shannon diversity (H') | 1.854±0.48 | 0.531±0.40 | $F_{1,4} = 13.52, P = 0.021$* |

Metrics of host-parasitoid food webs include LL: Species number of lepidopteran larvae in the network, HL: Species number of parasitoids in the network, LD: Linkage density, H': Shannon diversity

* $P < 0.05$.

**Table 6. Plot characteristics and abundance of *Fornicia* sp and lepidopteran larvae collected from the fogging method in Central Kalimantan.**

| No | Plot | Tree age (year) | Understorey vegetation | | Lepidopteran larvae | Parasitoid | |
| --- | --- | --- | --- | --- | --- | --- | --- |
| | | | Species | Cover (%) | | All | *Fornicia* sp |
| 1. | P121 | 12 | 19 | 42.6 | 775 | 201 | 14 |
| 2. | P122 | 12 | 17 | 40.0 | 365 | 223 | 8 |
| 3. | P141 | 14 | 20 | 90.9 | 605 | 140 | 12 |
| 4. | P142 | 14 | 18 | 66.5 | 383 | 235 | 5 |
| 5. | P161 | 16 | 22 | 70.7 | 453 | 294 | 7 |
| 6. | P162 | 16 | 18 | 44.5 | 485 | 138 | 9 |
| 7. | P171 | 17 | 14 | 36.7 | 444 | 235 | 10 |
| 8. | P172 | 17 | 18 | 100 | 309 | 258 | 7 |
| 9. | P181 | 18 | 17 | 20.4 | 811 | 264 | 26 |
| 10. | P182 | 18 | 20 | 57.6 | 599 | 207 | 6 |

Habitat conditions at the field scale, in this case, understorey vegetation, also did not affect the structure of host-parasitoid food webs in oil palm plantations. This might relate to the size and composition of understorey vegetation being insufficient to support parasitoid wasps. Our previous research also revealed that habitat conditions within and surrounding oil palm fields did not affect the species richness and abundance of parasitoid wasps [43]. This is surprising because understorey vegetation, such as flowering plants, in other agroecosystems can support parasitoids as alternative hosts and prey, alternative sources of pollen and nectar, shelter, favorable microclimates, or a combination of these resources [44,45]. At the field scale, flowering plants were reported to affect parasitoid diversity and food webs in agricultural landscapes. Research by [46] showed that species richness and abundance of natural enemies, as well as food web interactions, were particularly affected by the presence of flowers in an agroecosystem.

However, the diversity of parasitoids may also track changes in host diversity at each phase of crop plant growth [47]. Specialist parasitoid species such as *Fornicia* sp, *Aulosaphes* sp, and

**Table 7. Generalized linear models relating the abundance of lepidopteran larvae, all parasitoid wasps, and *Fornicia* sp to the tree age of oil palm, diversity and cover of understorey vegetation as predictors from ten plots of oil palm plantation in Central Kalimantan.**

| Variable | Estimate | SE | P |
| --- | --- | --- | --- |
| Lepidopteran larvae | | | |
| (Intercept) | 5.197 | 1.13 | 0.004 |
| Tree age | 0.008 | 0.04 | 0.859 |
| Vegetation diversity | 0.082 | 0.05 | 0.182 |
| Vegetation cover | -0.010 | 0.00 | 0.085 |
| All parasitoids | | | |
| (Intercept) | 4.860 | 1.04 | 0.003 |
| Tree age | 0.031 | 0.04 | 0.473 |
| Vegetation diversity | 0.005 | 0.05 | 0.919 |
| Vegetation cover | -0.001 | 0.00 | 0.889 |
| *Fornicia* sp | | | |
| (Intercept) | 2.097 | 2.01 | 0.336 |
| Tree age | 0.032 | 0.07 | 0.681 |
| Vegetation diversity | 0.029 | 0.10 | 0.768 |
| Vegetation cover | -0.015 | 0.01 | 0.154 |

*Elasmus* sp are always found in different tree ages of oil palm. *Fornicia* always occurs in different tree ages may be due to its host, *Darna trima*, and *D. deducta* (Lepidoptera: Limacodidae) also always occur in different tree ages of oil palm. The genus *Fornicia* Brullé (Braconidae: Microgastrinae) is a parasitoid of the Family Limacodidae [35], which is one of the important lepidopteran pests in oil palm plantation. While *Aulosaphes* sp and *Elasmus* sp are known to parasitize bagworms (Lepidoptera: Psychidae) [34]. *Aulosaphes* is a gregarious parasitoid, while *Elasmus* is a solitary parasitoid and hyperparasitic on bagworms [34].

This research also found that different geographical regions have different species compositions of lepidopteran pests and their parasitoids, as well as the structure of food webs. It means that each region has its environmental characteristics that shape the communities of pests and parasitoids in oil palm plantations. For instance, landscape characteristics also affect the structure of plant-insect food webs [48,49] as a consequence of the spill-over of natural enemies between the agroecosystem and natural ecosystem in the surrounding agroecosystem [50]. In Central Kalimantan, some remaining forests still exist surrounding plots of oil palm fields, while in Jambi, the plots of oil palm fields were located nearby other oil palm plantations or rubber plantations. The availability of alternative host species, shelters from agricultural disturbance, overwintering sites, and additional food sources provided by semi-natural habitats surrounding oil palm plantations may have contributed to the occurrence of parasitoids [51–54]. Although, in a particular condition, a semi-natural ecosystem can fail to support parasitoids in the agroecosystem [3].

These findings showed that besides oil palm plantation causing decreasing biodiversity [55,56], the long-term change of oil palm plantation, particularly increasing tree age of oil palm also causes decreasing the structure of host-parasitoid food webs. This is an anxious condition, as in the long term, the species composition and the species richness of communities may not be stable even if agricultural management does not change [24]. This condition can facilitate pestiferous species and give rise to pest outbreaks [57] in oil palm plantations. Therefore, managing habitat in oil palm plantations by spatiotemporal land-use diversification [58] or providing non-crop habitats in structurally complex landscapes may greatly enhance the activity and abundance of parasitoids [59,60] and increase the complexity and stability of host-parasitoid food webs [7].

## Conclusions

In conclusion, the change in host-parasitoid food webs in oil palm plantations over time tends to decrease with increasing age of the oil palm, highlighting the long-term effect of the monoculture system on biological control services in oil palm plantations. We also found that different geographical regions have a different patterns of change in host-parasitoid food webs. The differences in geographical regions suggest that habitat conditions, both at the field scale and landscape scale, may contribute to shaping the host-parasitoid interaction in oil palm plantations. Therefore, findings from other agroecosystems in different geographical regions, which suggest that agricultural landscape composition can affect host-parasitoid food webs [48,49], may also apply here. This suggests that managing oil palm plantations at a landscape scale can reduce the long-term effect of the monoculture system on biological control services. The relative role of landscape composition in supporting biological control in oil palm plantations will need to be investigated. The results will enhance our understanding of the characteristics of the oil palm plantation landscape that provide stability and function for conserving host-parasitoid food webs.

## Supporting information

**S1 Table. Raw data matrices of lepidopteran larvae and parasitoid wasps were generated based on a hand-collection method from 100 oil palm trees (aggregated over three**

**sampling events) in Central Kalimantan.** These matrices were used to calculate metrics for trophic interaction networks, which were analyzed using a bipartite ecological network approach (Dorman et al., 2009). Plot codes were derived from Table 1 in the text.
(DOCX)

**S2 Table. Raw data matrices of lepidopteran larvae and parasitoid wasps based on hand-collection method in 100 oil palm trees (aggregate 3 times sampling) in Jambi.** These matrices were used to obtain the metrics of trophic interaction networks, which were analyzed using a bipartite ecological network (Dorman et al., 2009).
(DOCX)

**S3 Table. List of parasitoid wasp morphospecies collected using the fogging method in an oil palm plantation in Central Kalimantan.** Plot codes were based on Table 6 in the text.
(DOCX)

## Acknowledgments

We thank PT Astra Agro Lestari, Tbk, for the permission to carry out field research in their oil palm plantations. We also thank Bandung Sahari, Nurindah, Radhian Ardy Prabowo, Ronny Pamuji, Ariatno, Sidik Purnomo, Santosa, and Agung Budianto, who provided their support during the research. We thank Prof. Fredy Kurniawan, who provided valuable comments during the workshop on manuscript preparation organized by the Ministry of Education, Culture, Research, and Technology year 2022. We would thank Prof. Lucas D. B. Faria, Prof. Edgar Turner, and anonymous reviewers who provided valuable inputs and comments to revise the manuscript.

## Author Contributions

**Conceptualization:** Akhmad Rizali, Sri Karindah, Bambang Tri Rahardjo, Damayanti Buchori.

**Data curation:** Nur Ainy, Muhamad Luthfie Tri Meiadi, Muhammad Iqbal Tawakkal.

**Formal analysis:** Akhmad Rizali.

**Funding acquisition:** Akhmad Rizali.

**Investigation:** Akhmad Rizali, Nur Ainy, Muhamad Luthfie Tri Meiadi, Muhammad Iqbal Tawakkal.

**Methodology:** Akhmad Rizali, Sri Karindah, Nur Ainy, Muhamad Luthfie Tri Meiadi, Muhammad Iqbal Tawakkal, Bambang Tri Rahardjo, Damayanti Buchori.

**Supervision:** Akhmad Rizali, Sri Karindah, Bambang Tri Rahardjo.

**Visualization:** Akhmad Rizali.

**Writing – original draft:** Akhmad Rizali, Sri Karindah, Bambang Tri Rahardjo, Damayanti Buchori.

**Writing – review & editing:** Akhmad Rizali.

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
