## [Decision Letter · Decision Letter 0]

23 Feb 2023

PONE-D-22-26542Long-term tree growth of oil palm simplifies the structure of host-parasitoid food websPLOS ONE

Dear Dr. Rizali,

Thank you for submitting your manuscript to PLOS ONE. After careful consideration, we feel that it has merit but does not fully meet PLOS ONE’s publication criteria as it currently stands. Therefore, we invite you to submit a revised version of the manuscript that addresses the points raised during the review process.

We look forward to receiving your revised manuscript.

Kind regards,

Lucas D. B. Faria

Academic Editor

PLOS ONE

“We thank PT Astra Agro Lestari, Tbk, for the permission to carry out field research in their oil palm plantations. We also thank Bandung Sahari, Nurindah, Radhian Ardy Prabowo, Ronny Pamuji, Ariatno, Sidik Purnomo, Santosa, and Agung Budianto, who provided their support during the research. We thank Prof. Fredy Kurniawan who provided valuable comments during the workshop on manuscript preparation, funded by the Ministry of Education, Culture, Research, and Technology year 2022.”

“AR received funding from the Indonesian Oil Palm Estate Fund Agency (BPDPKS), Ministry of Finance of the Republic of Indonesia year 2016. The funders had no role in study design, data collection and analysis, decision to publish, or preparation of the manuscript.”

5.We note that [Figure 1] in your submission contain [map/satellite] images which may be copyrighted. All PLOS content is published under the Creative Commons Attribution License (CC BY 4.0), which means that the manuscript, images, and Supporting Information files will be freely available online, and any third party is permitted to access, download, copy, distribute, and use these materials in any way, even commercially, with proper attribution. For these reasons, we cannot publish previously copyrighted maps or satellite images created using proprietary data, such as Google software (Google Maps, Street View, and Earth). For more information, see our copyright guidelines: http://journals.plos.org/plosone/s/licenses-and-copyright.

a. You may seek permission from the original copyright holder of Figure 1] to publish the content specifically under the CC BY 4.0 license. 

In the figure caption of the copyrighted figure, please include the following text: “Reprinted from [ref] under a CC BY license, with permission from [name of publisher], original copyright [original copyright year].

Natural Earth (public domain): http://www.naturalearthdata.com/.

Additional Editor Comments:

Revisions on the manuscript were made, and both reviewers pointed out relevant outcomes. Besides the comments and suggestions made by both referees, I also have some doubts about the statistical analysis. It is critical for the robustness of the results to understand the effects of different sampling methods on the interactions. As a result, I strongly advise assuming an accumulation curve based on the number of interactions between both methods. Is the effort enough to reach saturation for both methods? If not, how would it affect the outcomes? Please provide analysis and a clear incorporation of it into the main text. Further, GLM analyses were made; however, it is not clear why not GLMM (mixed effects) assuming plants as random variables in the models. Please provide a clear explanation about it, or change GLM for GLMM.

Reviewers' comments:

Reviewer's Responses to Questions

**Comments to the Author**

1. Is the manuscript technically sound, and do the data support the conclusions?

Reviewer #1: Yes

Reviewer #2: Partly

2. Has the statistical analysis been performed appropriately and rigorously? 

Reviewer #1: I Don't Know

Reviewer #2: N/A

3. Have the authors made all data underlying the findings in their manuscript fully available?

Reviewer #1: No

Reviewer #2: Yes

4. Is the manuscript presented in an intelligible fashion and written in standard English?

Reviewer #1: No

Reviewer #2: Yes

5. Review Comments to the Author

Reviewer #1: This interesting paper compares characteristics of lepidopteran and parasitoid species richness and food webs across different ages of oil palm, with sites in Central Borneo and Jambi Province, Sumatra. The study includes data from direct collection of larvae and raising of parasitoids from earlier ages of oil palm, and collections of larvae and parasitoids by insecticide fogging in later ages.

The work represents an enormous collection of data, providing a comprehensive overview of the effects of oil palm age on lepidopteran larvae, parasitoids, and their interactions. The key findings are that number of species of larvae, parasitoids and linkage density all decline with oil palm age in earlier years (from direct collection data), but this is not apparent in later years (from fogging data). This finding has important implications for understanding the ecology of oil palm systems and informing pest management approaches.

I felt that the study set-up was robust and the reporting fair, but I did find it quite hard to follow exactly how the study was carried out in places, particularly related to the analyses, and that the results could be discussed in more detail. I therefore recommend the changes below, which I hope are helpful. My overarching suggestions are:

1) I don’t think using the terms “tree growth” or “development” are clear, as this suggests it is just tree size that is changing – I would suggest rewording to “plantation age” throughout, to encompass all the other environmental differences that occur as plantations mature.

2) Please add more details to the methods about the two study sites and why these were chosen for comparison – this isn’t really clear at the moment.

3) Please give specific details of what factors were included in each analysis and whether assumptions of tests were assessed and met. I couldn’t really follow what was included in the different tests from the methods.

4) I wasn’t sure how Shannon was being calculated in this case (e.g. just for parasitoids?) Please add details to the methods.

5) I would like to know more about the effectiveness of raising larvae – e.g. what was the level of survival to adulthood or emergence of parasitoids across different species etc. and percentage of parasitism? Could differing survival across species impact food-web results?

6) I think the paper would also be improved by including some sort of compositional analysis, to quantify how larval and parasitoid composition changed across ages. Abundance of pests and parasitoids can be important, as well as diversity, so I also suggest including analyses on total abundances.

7) I didn’t find the discussion structure very easy to follow and I would recommend more discussion about the reasons for differences observed (e.g. what specific factors might be driving differences across ages?). I also suggest adding a specific section comparing the differing results between the hand collection and fogging studies.

I include specific points below:

Abstract:

The term “long-term tree development” isn’t very clear. Suggest replacing with “oil palm age”

It isn’t clear what the set-up is from the abstract and what ages were collected from Jambi – please give more details here.

“Results of the GLM analysis showed that the abundance of Fornicia sp was not affected by the tree age of oil palm but had a positive effect on the abundance of lepidopteran larvae.” Presumably, it is the number of larvae affecting the number of wasps rather than the other way round – please rewrite to reflect this.

Introduction:

60-61 – I don’t think it is clear what “harmonious and balanced interactions” are – suggest replacing with more specific wording.

65- I am not sure “relies on” is the right term here – suggest rewording to say that species and their interactions support ecosystem services.

71- Suggest rewording to “compare across ages”, rather than “growth of oil palm trees”, as many other factors are changing as well as height that could influence these interactions

73 – “more robust description” is this compared to descriptive methods? Please clarify.

83 – “imbalance” is a bit difficult to quantify – suggest rewording to something more specific such as “lower numbers of predators”, or “non-target effects on natural enemies”.

86-87 – It would be useful to include more information here about results from studies comparing differences in communities across different ages of oil palm.

93 – Suggest adding more justification here about why different regions included and chosen.

Methods:

97 – In this section, please add a description of the habitat conditions in both regions for context (e.g. climate, levels of natural vegetation, oil palm age structure across the landscape etc.)

103 -106 - Suggest moving mention of collection methods to later, after detailing the plot set-up. I think the term “visual observation” is a bit misleading, as it implies you just observed rather than collecting, perhaps use “hand-collection” instead?

108 – “randomly selected as sampling units” - suggest reword to “randomly selected within these plots as sampling units”

109-112 – It would be useful to add a justification here of why a comparison across areas was carried out?

121 – “within a hundred trees” - I’m not totally clear what was sampled here – were all 100 trees in each plot sampled for 15 minutes? Please clarify.

121 -123 – Suggest rewording to “We searched for lepidopteran larvae for a total of 15 minutes per tree, looking for larvae for 5 minutes in the top, middle, and bottom section of each canopy.” Presumably, you only searched the oil palm fronds themselves (ie not the epiphytes or anything else?)

130 – “feeding larvae.” – is this whether they fed between observations? Please clarify.

140 – Was the sheet placed right next to the tree (ie would it also collect insects from the trunk/epiphytes etc?)

147 – “diversity and cover of vegetation” please add more details – was it percentage cover by eye, number of all understory species (including grasses) etc.?

161 –Please clarify how Shannon diversity was calculated – was this done separately for parasitoids and larvae?

161-162 – Were data from all 100 trees in each plot pooled for all time periods, prior to analyses? Please give details.

166 – Please state what the specific dependent variables were. Presumably only Age 3 plantations were included for Central Borneo?

168-170 – Please clarify what dependent and independent variables were included in the models? E.g. was age included as well as environmental factors in single models? What aspects of parasitoids (abundance/diversity etc?) were included? Please state in full here. From looking at the results, I think this included number of lepidopteran species, number of parasitoid species, and aspects of foodweb structure. I think it would be good to test effects on total abundance. Please clarify that diagnostic tests were carried out to test assumptions of the tests.

Results:

179 -181 – This is really methods – please move to that section.

177-179 –Please also add summary stats on how many parasitoids raised, level of survival etc. It would be useful to also add a brief summary of the most common species.

184-185 – Please state the results more completely in the text. From looking at the table, only tree age comes out as being a significant predictor and it would be helpful to state this.

189 – “found in different tree ages” I’m not sure what you mean here – are these species found across all oil palm ages? Please clarify.

Figure 2 and associated text – It would be useful to know the number of larvae that weren’t parasitized, as well as survival of larva. If some species died more commonly, how likely is this to have affected your results?

Figure 3 – It would be helpful to add a community analysis (e.g. nMDS) to quantify differences in communities of larvae and wasps across ages.

208-209 – As above – it would be helpful to quantify how much this finding might be influenced by larval survival etc?

209 – 211 – Please give full results in text (including what doesn’t differ). Where a difference is present, please state the direction of difference.

219-220 – Were Fornicia spp. the only parasitoids found in these ages? Please clarify.

220-222 – Please move to methods.

Discussion

247-250 – I think it would be useful to discuss specifically what is it about the older palms that might lead to these differences?

254-256 – But only in the case of the fogging results. This result could reflect differing abilities to detect differences between techniques. Suggest adding a discussion point related to this. I think this lack of effect for older palms could also relate to mature palms all being rather similar environmentally once the canopy has closed, compared to differences in the environment between years in younger palms. Again, suggest adding a discussion point related to this.

261-271 – Why do you think that understory vegetation is not affecting parasitoids in this case? Please discuss.

282-283 – Although not directly tested in this study. As above, suggest carrying out multivariate tests to assess differences in composition.

285-292 – Do differences in landscape structure etc. between the two sites vary and in the direction you would predict from your results? Please add details.

I hope that these comments help to develop this important paper.

With all best wishes

Edgar Turner, Insect Ecology Group, Department of Zoology, University of Cambridge, UK.

Reviewer #2: The manuscript brings relevant information about how host-parasitoid food webs change with the increasing age of plants in oil palm, which can help in the ecological management of pests. The manuscript is well written, with clear and objective language. However, some information needs to be presented in the manuscript. I consider two main problems to be solved: 1 - The host-parasitoid interactions need to be reviewed and substantiated, as there are mistaken interactions. 2 - Both the number of species and the abundance of lepidopterans and parasitoids needs to be considered in the statistical models to make the results clearer.

MATERIALS AND METHODS

L120 Were the instars of the caterpillars considered in the collection or were the caterpillars collected indiscriminately? Make it clear.

L128 In the laboratory, were the insects raised in a controlled environment? Make it clear whether the photoperiod, humidity, and temperature conditions were controlled or natural.

L140 About the nebulization parasitoid collection method: how to guarantee that the dead parasitoids collected on the white canvas were associated with lepidopterans? The parasitoids could be visiting the tree.

L152 Lepidoptera and parasitoid species appear in tables in appendices S1 and S2 as morphospecies. It is necessary to mention whether all species were separated into morphospecies based on morphological characteristics.

L160 It should be made clear how the linkage density was calculated, as well as the Shannon diversity.

L169 Mention the packages used to build the models.

RESULTS

I have several questions about the collected parasitoids and their interactions:

Diapriidae.1. They are known to be parasitoids of larvae and pupae of several Diptera families, Coleoptera larvae such as Staphylinidae and Psephenidae families, and ant larvae. They are also recognized as hyperparasitoids through Dryinidae and Tachinidae. They are not associated with lepidopteran hosts (Please see Masner, 2006 Familia Diapriidae - in Hanson and Gauld, 2006 Hymenoptera de la Región Neotropical). Therefore, the parasitoid obtained certainly did not emerge from the lepidopteran Metisa plana, and could have emerged from a host different from the caterpillars collected. It could have been accidentally collected or be hyperparasitoid via Tachinidae, for example. In this case, it is necessary to exclude the species from the analyses.

Scelionidae.14: The parasitoids of this family are known to be idiobiont endoparasitoids of insect and spider eggs. The authors obtained 40 parasitoids that emerged from Lepidoptera.1. However, the methodology of this research was based on the collection of lepidopteran larvae from plants. How could the authors explain the presence of 40 egg parasitoids?

Chrysidae.sp1. They are known to be parasitoids of Hymenoptera (symphytes and nest builders) and Phasmatodea (Kimsey, 2006 Familia Chrysididae - in Hanson and Gauld, 2006 Hymenoptera de la Región Neotropical). Therefore, the parasitoids obtained certainly did not emerge from the lepidopteran Parasa lepida. In this case, as well as in the case of Diapriidae.1, it is necessary to exclude it from the analyses.

According to Noyes, 2019 (Universal Chalcidoidea Database) the Elasmidae family is currently considered as a tribe within Eulophinae (Eulophidae). Make corrections to tables and text.

As shown in the previous questions, host-parasitoid interactions are not scientifically supported. I understand that establishing these interactions with most parasitoids at the family taxonomic level is something very difficult. However, misinterpreted interactions compromise results in food web studies. I suggest that the authors prepare a scientifically based table for host-parasitoid interactions and place it in the Supporting Information.

In the tables in appendices S1 and S2, all parasitoids are identified at the family level. However, throughout the text, the authors pointed out some genera of the parasitoids obtained. It is necessary to add the identified parasitoid genera.

L 184. The authors mention that the increasing age of oil palm trees caused a decrease in the number of species of lepidopteran larvae and parasitoids. Subsequently, the authors showed that the abundance of lepidopteran larvae had a positive effect on the abundance of parasitoids. This suggests that the effect of increasing the age of trees on the decrease in parasitoid species may be an indirect effect mediated by the reduction in the abundance and number of Lepidoptera species. I believe that the authors should consider the abundance and number of lepidopteran and parasitoid species both in the models in which plant ages were tested and in the models in which parasitoid abundance was tested.

L 200 Fig 2 It is necessary to explain the size of the bars and links.

6. PLOS authors have the option to publish the peer review history of their article (what does this mean?). If published, this will include your full peer review and any attached files.

Reviewer #1: No

Reviewer #2: No

---

## [Author Response · Author response to Decision Letter 0]

30 Apr 2023

Dear Reviewers,

We would like to resubmit our manuscript ID PONE-D-22-26542 entitled “Long-term tree growth of oil palm simplifies the structure of host-parasitoid food webs" for publication in PLOS ONE.

We thank you for the constructive comments and we have revised the manuscript accordingly. In this version, we revised the title to “Long-term change of oil palm plantation simplifies the structure of host-parasitoid food webs”.

This document contains your comments and our responses are below the comments. Where we thought recommended changes were not appropriate, we give the reason in the response.

We hope you will find our manuscript suitable for publication in PLOS ONE.

Sincerely,

Akhmad Rizali

Reviewer #1:

This interesting paper compares characteristics of lepidopteran and parasitoid species richness and food webs across different ages of oil palm, with sites in Central Borneo and Jambi Province, Sumatra. The study includes data from direct collection of larvae and raising of parasitoids from earlier ages of oil palm, and collections of larvae and parasitoids by insecticide fogging in later ages.

The work represents an enormous collection of data, providing a comprehensive overview of the effects of oil palm age on lepidopteran larvae, parasitoids, and their interactions. The key findings are that number of species of larvae, parasitoids and linkage density all decline with oil palm age in earlier years (from direct collection data), but this is not apparent in later years (from fogging data). This finding has important implications for understanding the ecology of oil palm systems and informing pest management approaches.

I felt that the study set-up was robust and the reporting fair, but I did find it quite hard to follow exactly how the study was carried out in places, particularly related to the analyses, and that the results could be discussed in more detail. I therefore recommend the changes below, which I hope are helpful. My overarching suggestions are:

1) I don’t think using the terms “tree growth” or “development” are clear, as this suggests it is just tree size that is changing – I would suggest rewording to “plantation age” throughout, to encompass all the other environmental differences that occur as plantations mature.

We preferred to change the title with “Long-term change of oil palm plantation simplifies the structure of host-parasitoid food webs”. But in the whole manuscript, we have replaced the terms “tree growth” with “tree age” or “plantation age”.

2) Please add more details to the methods about the two study sites and why these were chosen for comparison – this isn’t really clear at the moment.

We have added the explanation in the methods “We selected Central Kalimantan and Jambi due to two regions have similar topography and climate that mainly lowland area and with tropical climates.”

3) Please give specific details of what factors were included in each analysis and whether assumptions of tests were assessed and met. I couldn’t really follow what was included in the different tests from the methods.

The main objective of this research is to study the effect of tree age and environment condition (understorey vegetations) on host-parasitoid food webs. We have added more detail about the factors in the method section

4) I wasn’t sure how Shannon was being calculated in this case (e.g. just for parasitoids?) Please add details to the methods.

To study the host-parasitoid food webs, we quantified using bipartite ecological networks that was developed by Dormann et al (2009, The Open Ecology Journal. 2:7-24). The bipartite analysis provided quantitative metrics from the food web, include of species number of LL (host), species number of HL (parasitoid), linkage density and Shannon diversity. Based on Dormann et al (2009), linkage density was calculated based on marginal totals-weighted diversity of interactions per species (quantitative) which is computed as the average of vulnerability and generality (Bersier et al. 2002). While Shannon diversity is Shannon's diversity of interactions (i.e., network entries).

We have added this explanation in the method section.

5) I would like to know more about the effectiveness of raising larvae – e.g. what was the level of survival to adulthood or emergence of parasitoids across different species etc. and percentage of parasitism? Could differing survival across species impact food-web results?

We have added in Table 2 and Table 4 to show the list of lepidopteran larvae and their parasitoids as well as the percentage of parasitism.

6) I think the paper would also be improved by including some sort of compositional analysis, to quantify how larval and parasitoid composition changed across ages. Abundance of pests and parasitoids can be important, as well as diversity, so I also suggest including analyses on total abundances.

We have added a table to show species richness and abundance of lepidopteran larvae and their parasitoids. We prefer not to analysis the species composition due to it has already show in the food-webs.

7) I didn’t find the discussion structure very easy to follow and I would recommend more discussion about the reasons for differences observed (e.g. what specific factors might be driving differences across ages?). I also suggest adding a specific section comparing the differing results between the hand collection and fogging studies.

This study used two different methods i.e., hand-collection method and the fogging method. The hand-collection method was carried out at 9 plots with each plot the observations were conducted in 100 oil palm trees with the purpose: (1) to confirm that the parasitoid wasps are associated with their hosts (lepidopteran pests) and (2) to ensure that samples can be obtained sufficient because each oil palm trees are not always found lepidoptera larvae. In addition, due to this method is only possible for oil palms under 10 years old, we also conducted fogging method to get the information about parasitoid wasps and lepidopteran larvae from oil palm trees above 10 years old. This method was carried out on different plots/locations from hand-collection method i.e., in 10 plots with each plot selected 6 oil palm trees as sample units. However, the host-parasitoid food webs can only be analysed based on the data from hand-collection method.

I include specific points below:

Abstract:

The term “long-term tree development” isn’t very clear. Suggest replacing with “oil palm age” It isn’t clear what the set-up is from the abstract and what ages were collected from Jambi – please give more details here.

We have replaced the term “long-term tree development” with “long-term change of oil palm plantation” and we have also added the tree age of oil palm in Jambi

“Results of the GLM analysis showed that the abundance of Fornicia sp was not affected by the tree age of oil palm but had a positive effect on the abundance of lepidopteran larvae.” Presumably, it is the number of larvae affecting the number of wasps rather than the other way round – please rewrite to reflect this.

We have reanalysed the data and rewritten the sentence

Introduction:

60-61 – I don’t think it is clear what “harmonious and balanced interactions” are – suggest replacing with more specific wording.

We preferred to use “balance interaction” and delete “harmonious”

65- I am not sure “relies on” is the right term here – suggest rewording to say that species and their interactions support ecosystem services.

We have reworded with “The importance of conserving these interactions and associated processes, as well as the component species that associated with species interactions, such as pollination and biological control, supports ecosystem services"

71- Suggest rewording to “compare across ages”, rather than “growth of oil palm trees”, as many other factors are changing as well as height that could influence these interactions

We rephrased the sentence to “This paper investigates how the long-term change of oil palm plantation, specifically the tree age, effects host-parasitoid food webs”

73 – “more robust description” is this compared to descriptive methods? Please clarify.

No, this is not compared. We replaced to “more robust quantification”

83 – “imbalance” is a bit difficult to quantify – suggest rewording to something more specific such as “lower numbers of predators”, or “non-target effects on natural enemies”.

We have replaced to “non-target effects on natural enemies”

86-87 – It would be useful to include more information here about results from studies comparing differences in communities across different ages of oil palm.

We have added the information about the effect of oil palm tree age on insect communities

93 – Suggest adding more justification here about why different regions included and chosen.

We have added a sentence as justification

Methods:

97 – In this section, please add a description of the habitat conditions in both regions for context (e.g. climate, levels of natural vegetation, oil palm age structure across the landscape etc.)

We have added the description of climate

103 -106 - Suggest moving mention of collection methods to later, after detailing the plot set-up. I think the term “visual observation” is a bit misleading, as it implies you just observed rather than collecting, perhaps use “hand-collection” instead?

We have moved the collection methods in the next subheading and change the term “visual observation” with “hand-collection”

108 – “randomly selected as sampling units” - suggest reword to “randomly selected within these plots as sampling units”

We have revised the sentence

109-112 – It would be useful to add a justification here of why a comparison across areas was carried out?

We have added the explanation and justification in previous paragraph

121 – “within a hundred trees” - I’m not totally clear what was sampled here – were all 100 trees in each plot sampled for 15 minutes? Please clarify.

We have revised the sentence by replace with “...for all hundred trees” and explain the sampling time in the next sentence

121 -123 – Suggest rewording to “We searched for lepidopteran larvae for a total of 15 minutes per tree, looking for larvae for 5 minutes in the top, middle, and bottom section of each canopy.” Presumably, you only searched the oil palm fronds themselves (ie not the epiphytes or anything else?)

Yes, we only searched the oil palm canopy without epiphytes. We have rephrased the sentence 

130 – “feeding larvae.” – is this whether they fed between observations? Please clarify.

We decided to delete this word

140 – Was the sheet placed right next to the tree (ie would it also collect insects from the

trunk/epiphytes etc?)

Yes, it was. Because it was difficult to avoid the kill insects from the trunk/epiphytes

147 – “diversity and cover of vegetation” please add more details – was it percentage cover by eye, number of all understory species (including grasses) etc.?

Yes, it was. We have added the description in the next sentence

161 –Please clarify how Shannon diversity was calculated – was this done separately for parasitoids and larvae?

To study the host-parasitoid food webs, we quantified using bipartite ecological networks that was developed by Dormann et al (2009, The Open Ecology Journal. 2:7-24). The bipartite analysis provided quantitative metrics from the food web, include of species number of LL (host), species number of HL (parasitoid), linkage density and Shannon diversity. Based on Dormann et al (2009), linkage density was calculated based on marginal totals-weighted diversity of interactions per species (quantitative) which is computed as the average of vulnerability and generality (Bersier et al. 2002). While Shannon diversity is Shannon's diversity of interactions (i.e., network entries).

We have added this explanation in the method section.

161-162 – Were data from all 100 trees in each plot pooled for all time periods, prior to analyses? Please give details.

Due to, the data per time period is limited. We pooled the data in each plot for all time periods prior to analysis

166 – Please state what the specific dependent variables were. Presumably only Age 3 plantations were included for Central Borneo?

We reanalysed using ANOVA and add the explanation about responses and explanatory variables which were used.

168-170 – Please clarify what dependent and independent variables were included in the models? E.g. was age included as well as environmental factors in single models? What aspects of parasitoids (abundance/diversity etc?) were included? Please state in full here. From looking at the results, I think this included number of lepidopteran species, number of parasitoid species, and aspects of foodweb structure. I think it would be good to test effects on total abundance. Please clarify that diagnostic tests were carried out to test assumptions of the tests.

We have added the responses and explanatory variables as well as the explanation about model assumption.

Results:

179 -181 – This is really methods – please move to that section.

We have moved to the methods section

177-179 –Please also add summary stats on how many parasitoids raised, level of survival etc. It would be useful to also add a brief summary of the most common species.

We have added the summary of parasitoids emerged and the parasitization level of each parasitoid

184-185 – Please state the results more completely in the text. From looking at the table, only tree age comes out as being a significant predictor and it would be helpful to state this.

We have added more description about the analysis results of GLM

189 – “found in different tree ages” I’m not sure what you mean here – are these species found across all oil palm ages? Please clarify.

Yes, these species were found across all oil palm ages. We have revised the sentences

Figure 2 and associated text – It would be useful to know the number of larvae that weren’t parasitized, as well as survival of larva. If some species died more commonly, how likely is this to have affected your results?

Based on the bipartite analysis (Dormann et al 2009), the data that were used for analysis were individual number of parasitoids emerged from each host (lepidopteran pest). We have added Table 2 to showed the detail of parasitisation level of each parasitoid. 

Figure 3 – It would be helpful to add a community analysis (e.g. nMDS) to quantify differences in communities of larvae and wasps across ages.

We prefer to focus on bipartite analysis as a tool to quantify the ecological networks that had been used another researcher such as Tylianakis et al 2007, Nature 445:202-205. With the metric can show the structure of host-parasitoid food webs.

208-209 – As above – it would be helpful to quantify how much this finding might be influenced by larval survival etc?

We have added Table 4 to show the parasitization level of each parasitoid

209 – 211 – Please give full results in text (including what doesn’t differ). Where a difference is present, please state the direction of difference.

We have added more description in the text

219-220 – Were Fornicia spp. the only parasitoids found in these ages? Please clarify.

We have added a short description of parasitoid diversity from fogging method

220-222 – Please move to methods.

We have moved to method section

Discussion

247-250 – I think it would be useful to discuss specifically what is it about the older palms that might lead to these differences?

We have added a discussion sentence about the consequence of oil palm tree growth

254-256 – But only in the case of the fogging results. This result could reflect differing abilities to detect differences between techniques. Suggest adding a discussion point related to this. I think this lack of effect for older palms could also relate to mature palms all being rather similar environmentally once the canopy has closed, compared to differences in the environment between years in younger palms. Again, suggest adding a discussion point related to this.

We have added a discussion related to this

261-271 – Why do you think that understory vegetation is not affecting parasitoids in this case? Please discuss.

We have added a discussion

282-283 – Although not directly tested in this study. As above, suggest carrying out multivariate tests to assess differences in composition.

In this research, we are only focus on host-parasitoid food webs. And related to the species composition, we have shown in Table 2 and Table 4

285-292 – Do differences in landscape structure etc. between the two sites vary and in the direction you would predict from your results? Please add details.

We have added a description about the two regions in the methodology. And in this research, we did not analyse the landscape structure and only predict based on previous research.

I hope that these comments help to develop this important paper.

Thank you very much for the inputs and comments

With all best wishes

Edgar Turner, Insect Ecology Group, Department of Zoology, University of Cambridge, UK.

 

Reviewer #2:

The manuscript brings relevant information about how host-parasitoid food webs change with the increasing age of plants in oil palm, which can help in the ecological management of pests.

The manuscript is well written, with clear and objective language. However, some information needs to be presented in the manuscript. I consider two main problems to be solved:

1 - The host-parasitoid interactions need to be reviewed and substantiated, as there are mistaken interactions.

In this research, we only used the data of hand-collection method to develop host-parasitoid interaction. We have made sure that the parasitoid wasps emerged from the collected hosts (lepidopteran larvae)

2 - Both the number of species and the abundance of lepidopterans and parasitoids needs to be considered in the statistical models to make the results clearer.

We have revised the Table 1 and added more detail about abundance and species richness of hosts and parasitoids. Due to we only focused on the structure of food webs, we only used the metric of food webs analysis based on bipartite (we have described in the method section). In this case, the response variables are LL: species number of lepidopteran larvae in the network, HH: species number of parasitoids in the network, LD: linkage density, H’: Shannon diversity

MATERIALS AND METHODS

L120 Were the instars of the caterpillars considered in the collection or were the caterpillars collected indiscriminately? Make it clear.

All instars of lepidopteran larvae were collected. We have added more description to make it clearer

L128 In the laboratory, were the insects raised in a controlled environment? Make it clear whether the photoperiod, humidity, and temperature conditions were controlled or natural.

No, lepidopteran larvae were not reared in a controlled environment. We have added more description

L140 About the nebulization parasitoid collection method: how to guarantee that the dead parasitoids collected on the white canvas were associated with lepidopterans? The parasitoids could be visiting the tree.

With the fogging method, we could not make sure that the parasitoids were associated with lepidopteran larvae. Therefore, we only focus on species Fornicia sp (as case study) that has been known (based on hand-collection method) attack lepidopteran larvae and always found in all fogging plots (tree age range from 12 to 18 years old).

L152 Lepidoptera and parasitoid species appear in tables in appendices S1 and S2 as morphospecies. It is necessary to mention whether all species were separated into morphospecies based on morphological characteristics.

We have added a description about the identification process

L160 It should be made clear how the linkage density was calculated, as well as the Shannon diversity.

To study the host-parasitoid food webs, we quantified using bipartite ecological networks that was developed by Dormann et al (2009, The Open Ecology Journal. 2:7-24). The bipartite analysis provided quantitative metrics from the food web, include of species number of LL (host), species number of HL (parasitoid), linkage density and Shannon diversity. Based on Dormann et al (2009), linkage density was calculated based on marginal totals-weighted diversity of interactions per species (quantitative) which is computed as the average of vulnerability and generality (Bersier et al. 2002). While Shannon diversity is Shannon's diversity of interactions (i.e., network entries).

We have added this explanation in the method section.

L169 Mention the packages used to build the models.

We did not use the package for GLM analysis. We only used the package of bipartite to analyse the metric of host parasitoid food webs and we have described in the method section.

RESULTS

I have several questions about the collected parasitoids and their interactions:

Diapriidae.1. They are known to be parasitoids of larvae and pupae of several Diptera families, Coleoptera larvae such as Staphylinidae and Psephenidae families, and ant larvae. They are also recognized as hyperparasitoids through Dryinidae and Tachinidae. They are not associated with lepidopteran hosts (Please see Masner, 2006 Familia Diapriidae - in Hanson and Gauld, 2006 Hymenoptera de la Región Neotropical). Therefore, the parasitoid obtained certainly did not emerge from the lepidopteran Metisa plana, and could have emerged from a host different from the caterpillars collected. It could have been accidentally collected or be hyperparasitoid via Tachinidae, for example. In this case, it is necessary to exclude the species from the analyses.

Thank you for the correction. We have excluded this morphospecies

Scelionidae.14: The parasitoids of this family are known to be idiobiont endoparasitoids of insect and spider eggs. The authors obtained 40 parasitoids that emerged from Lepidoptera.1. However, the methodology of this research was based on the collection of lepidopteran larvae from plants. How could the authors explain the presence of 40 egg parasitoids?

We decided to excluded this morphospecies too

Chrysidae.sp1. They are known to be parasitoids of Hymenoptera (symphytes and nest builders) and Phasmatodea (Kimsey, 2006 Familia Chrysididae - in Hanson and Gauld, 2006 Hymenoptera de la Región Neotropical). Therefore, the parasitoids obtained certainly did not emerge from the lepidopteran Parasa lepida. In this case, as well as in the case of Diapriidae.1, it is necessary to exclude it from the analyses.

Thank you for the correction. We have excluded this morphospecies

According to Noyes, 2019 (Universal Chalcidoidea Database) the Elasmidae family is currently considered as a tribe within Eulophinae (Eulophidae). Make corrections to tables and text.

We have replaced Elasmidae to Eulophinae

As shown in the previous questions, host-parasitoid interactions are not scientifically supported. I understand that establishing these interactions with most parasitoids at the family taxonomic level is something very difficult. However, misinterpreted interactions compromise results in food web studies.

I suggest that the authors prepare a scientifically based table for host-parasitoid interactions and place it in the Supporting Information.

We have added the tables in the Supporting Information as well as added Table 2 and Table 4 to show the list of lepidopteran pests and their parasitoids

In the tables in appendices S1 and S2, all parasitoids are identified at the family level. However,

throughout the text, the authors pointed out some genera of the parasitoids obtained. It is necessary to add the identified parasitoid genera.

If possible, some morphospecies were identified to genera level. We have added the description in the method section as well as replaced the family name to genera name in the appendices

L 184. The authors mention that the increasing age of oil palm trees caused a decrease in the number of species of lepidopteran larvae and parasitoids. Subsequently, the authors showed that the abundance of lepidopteran larvae had a positive effect on the abundance of parasitoids. This suggests that the effect of increasing the age of trees on the decrease in parasitoid species may be an indirect effect mediated by the reduction in the abundance and number of Lepidoptera species. I believe that the authors should consider the abundance and number of lepidopteran and parasitoid species both in the models in which plant ages were tested and in the models in which parasitoid abundance was tested.

We have reanalysed the data to provide the similar model with the data of hand-collection methods

L 200 Fig 2 It is necessary to explain the size of the bars and links.

We have added more description to explain the interaction graph

---

## [Decision Letter · Decision Letter 1]

19 Jun 2023

PONE-D-22-26542R1Long-term change of oil palm plantation simplifies the structure of host-parasitoid food websPLOS ONE

Dear Dr. Rizali,

Thank you for submitting your manuscript to PLOS ONE. After careful consideration, we feel that it has merit but does not fully meet PLOS ONE’s publication criteria as it currently stands. Therefore, we invite you to submit a revised version of the manuscript that addresses the points raised during the review process.

We look forward to receiving your revised manuscript.

Kind regards,

Lucas D. B. Faria

Academic Editor

PLOS ONE

Reviewers' comments:

Reviewer's Responses to Questions

**Comments to the Author**

1. If the authors have adequately addressed your comments raised in a previous round of review and you feel that this manuscript is now acceptable for publication, you may indicate that here to bypass the “Comments to the Author” section, enter your conflict of interest statement in the “Confidential to Editor” section, and submit your "Accept" recommendation.

Reviewer #1: (No Response)

Reviewer #2: All comments have been addressed

2. Is the manuscript technically sound, and do the data support the conclusions?

Reviewer #1: Yes

Reviewer #2: Yes

3. Has the statistical analysis been performed appropriately and rigorously? 

Reviewer #1: Yes

Reviewer #2: Yes

4. Have the authors made all data underlying the findings in their manuscript fully available?

Reviewer #1: No

Reviewer #2: Yes

5. Is the manuscript presented in an intelligible fashion and written in standard English?

Reviewer #1: Yes

Reviewer #2: Yes

6. Review Comments to the Author

Reviewer #1: This interesting and data-rich manuscript is much clearer and most of my previous comments have been well-addressed. However, I still think that some sections would benefit from more clarifying, and I have some additional suggestions that I hope will help to improve the manuscript.

1) It was really helpful to see the abundance of Lepidoptera and parasitoids included in the table for section one (the hand collection), but it would be really useful to analyse changes in abundance through a GLM framework as well – it feels like this is missing from the manuscript at the moment. This is important, especially as the total number of larvae may have a bigger effect on herbivory than number of larval species, and I recommend this analysis is included. I can see why it may not make sense to do this for parasitoids, as these were raised from the larvae, and therefore may not reflect genuine abundance in the field (it would still be useful to state why this isn’t analysed in the text of the methods).

2) Related to this, I would also recommend analysing number of larval species (if available?) in section two of the study (the fogging). I think this would be really helpful, as it would mean that the same factors, as far as possible, are being analysed against habitat characteristics in both sections. If larvae weren’t identified to species because of limited time or other constraints, then this could be stated in the methods.

3) I don’t think is it clear enough why Fornicia was focussed on (rather than all parasitoids) in the second section. I also wasn’t completely sure whether it was Fornicia abundance or number of species (I think the latter) that was analysed? I think this should be clarified in the methods and results. In this case, I think abundance could also be analysed, as the fogging will be picking up a reasonable estimation of the number of parasitoids present. If only Fornicia were identified for time/pragmatic reasons, then I recommend this is stated in the methods.

4) I am still unclear how successful raising the larvae was and how many died (but not as a result of parasitization)? I know from experience that this can be quite high (often due to fungal infections and things) and it would be useful to have this information in the paper, if available. Related to this, was % parasitism calculated as “number of larvae with parasitoids emerging/number of larvae that successfully emerged to adulthood or had parasitoids emerged” or “number of larvae with parasitoids emerging/total number of larvae brought back” I guess the former, but I think this should be explicitly stated, as it could make quite a big difference to the percentage.

5) Finally, I recommend that the negative relationship between understory cover and lepidopteran larval species richness is discussed in the discussion. It would be good to add this as a short paragraph, as it has interesting management implications. Related to this, it would be useful to discuss briefly why number of understory plant species did not affect anything.

Other than that, my specific comments are:

Title – I still don’t feel this is quite clear enough at first read. Suggest tweak wording to “Long-term changes as oil palm plantations age simplify the structure of host-parasitoid food webs”?

Abstract.

35 – “Shannon diversity” I think non-specialists will think of simple diversity indices for each group when reading this, as I did. To clarify, I suggest you reword to “Interaction diversity”, clarifying that this was calculated through Shannon's in the methods. I think this will be clearer for a wider readership.

39-41 – “Different geographical regions showed different host parasitoid food webs structure, especially species number of lepidopteran larvae and Shannon diversity.” please tweak the wording to say in which direction number of species and Shannon diversity differed.

41-43 – “Based on fogging method, Fornicia sp (Hymenoptera: Braconidae) were recorded in a different tree age range from 12 to 18 years.” This isn’t quite clear – suggest rewording to “For example, Fornicia sp (Hymenoptera: Braconidae) were recorded in all ages of oil palm sampled.” As I think them being found across a range of ages is the key point here?

64 – I’m not completely sure what “balanced interactions” are – can you be more specific in the terminology here?

Introduction

82-83 – As above, suggest specifying how Shannon’s used in this case.

Methods

119-120 – I think that “and without observation of parasitoid wasp diversity and lepidopteran larvae abundance.” Makes it a bit confusing what was measured. Suggest stating what was measured instead, perhaps “In this area, we selected three plots with trees aged three years and recorded host-parasitoid food webs, using the same methods applied in Central Kalimantan.”

130 – “Borneo” – you use Kalimantan, Central Kalimantan, Central Borneo, and Borneo at different points in the manuscript. Suggest simplifying to “Kalimantan” throughout for clarity (especially for those not familiar with the region).

133 – “with natural environmental conditions” –reword to “at room temperature” for clarity, as other conditions very different from the field.

151-152 – As the sheet was next to the tree, suggest adding “As the sheet was next to the tree, collections would likely include insects collected from both the oil palm canopy and from epiphytes growing on the trunk”, as a small caveat.

161-162 – Suggest adding a line to clarify what data was generated from this species identification. Perhaps: “From this we calculated number of understory species per quadrat” or something like that?

175 – As earlier, suggest you give more details here of exactly how linkage density and Shannon diversity are calculated in this case, for non-experts.

185 – Suggest rewording “With assumed” to “Assuming”

189 – By “diversity” do you mean number of species of understory plants? Please tweak wording to clarify.

192 “from the data of fogging method, we studied the effect of explanatory variables on Fornicia sp and lepidopteran abundance.” – As these variables were calculated from the raw data, rather than from the outputs of the bipartite ecological network, I think this should be moved to the end of the “Insect identification” section, above where you talk about the bipartite ecological network. Perhaps something like “From the fogging data, we calculated Lepidoptera larval abundance and the number of Fornicia sp per plot” Was this carried out per plot or per tree? I wasn’t sure whether it was Fornicia sp number of species or abundance calculated? Please clarify. In the section “Observation of Parasitoid Wasp Diversity and Lepidopteran Larvae Abundance”, I think you need to say why data on Fornicia only used, rather than all parasitoids. Presumably, this is because this was the commonest group and found in reasonable abundance, so you are using this as an indicator of parasitoids? Please state in the methods to make sure it is clear.

195 – “metric of networks” - for clarity, suggest you specifically state the metrics compared here again.

Results

206 – 208 – suggest adding overall parasitisation rates here as well.

211 “lepidopteran larvae was also affected by cover of understorey vegetation” please say in which direction this relationship was (negative)

228 – Table 2 legend – include mention of parasitisation level in the legend, and briefly say how this was calculated (% of individual larvae with parasitoids emerging per species?). I am not sure whether this is calculated for all larvae or just larvae that survived to pupation/had parasitoid emergence – I think this should be clarified. Please also include mention of total number of plots represented in the legend, to allow the table to be interpreted without referring back to the text

Table 2 – Please write abbreviations like “No. par” in full to avoid confusion

232 – Table 3 legend – as above, please give details of number of plots in the legend.

236-243 – As above, give details of number of plots represented in the figure legends

249-250 – Rather than just significantly different, state which is higher/lower in terms of Lepidoptera species number and Shannon diversity in Jambi versus Kalimantan for clarity.

253 -254 – Table 4 legend – Please give details of the number of plots in the legend, as above for other tables. Write abbreviations like “No. par” in full to avoid confusion.

257 – Table 5 legend – please give details of number of plots in the legend.

270-272 – As above – please say in which direction this marginal effect is (ie it is a negative association).

262-277 – Section “Effect of host abundance, tree age, and understorey vegetation on parasitoid abundance”. As above, I am not totally sure whether it is number of Fornicia species or abundance– I think the former. Please clarify.

278 – Table 7 legend – as above, please give N values in legend.

Discussion

209 -309 – Suggest adding a line or two to the end of this section, stating that the different results between the two sections of the study could be due to different methods and metrics used, with direct collection and raising of parasitoids likely to be more sensitive to the effects of oil palm age.

310-311 – “understorey vegetation, also did not affect the structure of host-parasitoid food webs in oil palm plantations.” Although number of larvae species did decrease with more cover? I think this should be added in here and discussed briefly, as it has interesting management implications.

332-342 – It would be useful to add some brief details of what the two study regions were like in the methods – ie did they differ in amount of forest cover etc, as you suggest could be driving results here?

Conclusion

355-368 – I think the findings also suggest that it might be best to replant oil palm in smaller blocks, creating a more diverse age-structure in oil palms?

I hope that these additional comments help with this interesting and important paper.

Edgar Turner, Insect Ecology Group, Department of Zoology, University of Cambridge.

Reviewer #2: (No Response)

7. PLOS authors have the option to publish the peer review history of their article (what does this mean?). If published, this will include your full peer review and any attached files.

Reviewer #1: **Yes: **Edgar Turner

Reviewer #2: No

---

## [Author Response · Author response to Decision Letter 1]

27 Jul 2023

Dear Reviewer,

We would like to resubmit our manuscript ID PONE-D-22-26542R1 entitled “Long-term tree growth of oil palm simplifies the structure of host-parasitoid food webs" for publication in PLOS ONE.

We thank you for the constructive comments and we have revised the manuscript accordingly. In this version, we revised the title to “Long-term changes as oil palm plantation age simplify the structure of host-parasitoid food webs”.

This letter contains your comments and our response directly below your comments. Where we thought recommended changes were not appropriate, we give the reason in the response.

We hope you will find our manuscript suitable for publication in PLOS ONE.

Sincerely,

Akhmad Rizali

Reviewer #1:

This interesting and data-rich manuscript is much clearer and most of my previous comments have been well-addressed. However, I still think that some sections would benefit from more clarifying, and I have some additional suggestions that I hope will help to improve the manuscript.

1) It was really helpful to see the abundance of Lepidoptera and parasitoids included in the table for section one (the hand collection), but it would be really useful to analyse changes in abundance through a GLM framework as well – it feels like this is missing from the manuscript at the moment. This is important, especially as the total number of larvae may have a bigger effect on herbivory than number of larval species, and I recommend this analysis is included. I can see why it may not make sense to do this for parasitoids, as these were raised from the larvae, and therefore may not reflect genuine abundance in the field (it would still be useful to state why this isn’t analysed in the text of the methods).

We have added the GLM result of lepidopteran larvae abundance in the Table 3

2) Related to this, I would also recommend analysing number of larval species (if available?) in section two of the study (the fogging). I think this would be really helpful, as it would mean that the same factors, as far as possible, are being analysed against habitat characteristics in both sections. If larvae weren’t identified to species because of limited time or other constraints, then this could be stated in the methods.

Thank you for the recommendation. Unfortunately, we did not identify the specimens of lepidopteran larvae to species level due to very difficult and need long time to be conducted.

3) I don’t think is it clear enough why Fornicia was focussed on (rather than all parasitoids) in the second section. I also wasn’t completely sure whether it was Fornicia abundance or number of species (I think the later) that was analysed? I think this should be clarified in the methods and results. In this case, I think abundance could also be analysed, as the fogging will be picking up a reasonable estimation of the number of parasitoids present. If only Fornicia were identified for time/pragmatic reasons, then I recommend this is stated in the methods.

We have reanalysed the data with included the abundance of all parasitoid wasps. And in the methodology, we also have added the reason why Fornicia was also included in the analysis.

4) I am still unclear how successful raising the larvae was and how many died (but not as a result of parasitization)? I know from experience that this can be quite high (often due to fungal infections and things) and it would be useful to have this information in the paper, if available. Related to this, was % parasitism calculated as “number of larvae with parasitoids emerging/number of larvae that successfully emerged to adulthood or had parasitoids emerged” or “number of larvae with parasitoids emerging/total number of larvae brought back” I guess the former, but I think this should be explicitly stated, as it could make quite a big difference to the percentage.

We have added the data of (i) number of collected larvae and (ii) number of % parasitized larvae in Table 2. In this research, we did not observe other factors that affecting the died of lepidopteran larvae during the rearing in the laboratory, except the died because of parasitized by parasitoids.

5) Finally, I recommend that the negative relationship between understory cover and lepidopteran larval species richness is discussed in the discussion. It would be good to add this as a short paragraph, as it has interesting management implications. Related to this, it would be useful to discuss briefly why number of understory plant species did not affect anything.

We reanalysed the data and we found understorey vegetation are not significant both in hand-collecting method (P=0.090) and fogging method (P=0.085)

Other than that, my specific comments are:

Title – I still don’t feel this is quite clear enough at first read. Suggest tweak wording to “Long-term changes as oil palm plantations age simplify the structure of host-parasitoid food webs”?

We have revised the title

Abstract.

35 – “Shannon diversity” I think non-specialists will think of simple diversity indices for each group when reading this, as I did. To clarify, I suggest you reword to “Interaction diversity”, clarifying that this was calculated through Shannon's in the methods. I think this will be clearer for a wider readership.

We agree to reword “Shannon diversity” to “interaction diversity”

39-41 – “Different geographical regions showed different host parasitoid food webs structure, especially species number of lepidopteran larvae and Shannon diversity.” please tweak the wording to say in which direction number of species and Shannon diversity differed.

We have revised the sentence

41-43 – “Based on fogging method, Fornicia sp (Hymenoptera: Braconidae) were recorded in a different tree age range from 12 to 18 years.” This isn’t quite clear – suggest rewording to “For example, Fornicia sp (Hymenoptera: Braconidae) were recorded in all ages of oil palm sampled.” As I think them being found across a range of ages is the key point here?

We have reworded

64 – I’m not completely sure what “balanced interactions” are – can you be more specific in the terminology here?

We have revised to “stability of interaction networks”

Introduction

82-83 – As above, suggest specifying how Shannon’s used in this case.

We have reworded to interaction diversity

Methods

119-120 – I think that “and without observation of parasitoid wasp diversity and lepidopteran larvae abundance.” Makes it a bit confusing what was measured. Suggest stating what was measured instead, perhaps “In this area, we selected three plots with trees aged three years and recorded host-parasitoid food webs, using the same methods applied in Central Kalimantan.”

We have revised the sentence

130 – “Borneo” – you use Kalimantan, Central Kalimantan, Central Borneo, and Borneo at different points in the manuscript. Suggest simplifying to “Kalimantan” throughout for clarity (especially for those not familiar with the region).

We agree to change Borneo to Kalimantan

133 – “with natural environmental conditions” –reword to “at room temperature” for clarity, as other conditions very different from the field.

We have reworded

151-152 – As the sheet was next to the tree, suggest adding “As the sheet was next to the tree, collections would likely include insects collected from both the oil palm canopy and from epiphytes growing on the trunk”, as a small caveat.

We have added the sentence

161-162 – Suggest adding a line to clarify what data was generated from this species identification. Perhaps: “From this we calculated number of understory species per quadrat” or something like that?

We have revised the sentence

175 – As earlier, suggest you give more details here of exactly how linkage density and Shannon diversity are calculated in this case, for non-experts.

We have added the description

185 – Suggest rewording “With assumed” to “Assuming”

We have reworded

189 – By “diversity” do you mean number of species of understory plants? Please tweak wording to clarify.

We have reworded

192 “from the data of fogging method, we studied the effect of explanatory variables on Fornicia sp and lepidopteran abundance.” – As these variables were calculated from the raw data, rather than from the outputs of the bipartite ecological network, I think this should be moved to the end of the “Insect identification” section, above where you talk about the bipartite ecological network. Perhaps something like “From the fogging data, we calculated Lepidoptera larval abundance and the number of Fornicia sp per plot” Was this carried out per plot or per tree? I wasn’t sure whether it was Fornicia sp number of species or abundance calculated? Please clarify. In the section “Observation of Parasitoid Wasp Diversity and Lepidopteran Larvae Abundance”, I think you need to say why data on Fornicia only used, rather than all parasitoids. Presumably, this is because this was the commonest group and found in reasonable abundance, so you are using this as an indicator of parasitoids? Please state in the methods to make sure it is clear.

We have moved the sentence in the suggested section. And in this revised version, we included the abundance of all parasitoid wasps in the analysis, beside the abundance of Fornicia sp.

195 – “metric of networks” - for clarity, suggest you specifically state the metrics compared here again.

We have added the detail of network metrics

Results

206 – 208 – suggest adding overall parasitisation rates here as well.

We have added in the previous sentence

211 “lepidopteran larvae was also affected by cover of understorey vegetation” please say in which direction this relationship was (negative)

We have revised the sentence

228 – Table 2 legend – include mention of parasitisation level in the legend, and briefly say how this was calculated (% of individual larvae with parasitoids emerging per species?). I am not sure whether this is calculated for all larvae or just larvae that survived to pupation/had parasitoid emergence – I think this should be clarified. Please also include mention of total number of plots represented in the legend, to allow the table to be interpreted without referring back to the text

We have added the detail in the legend

Table 2 – Please write abbreviations like “No. par” in full to avoid confusion

We have written in full in the table

232 – Table 3 legend – as above, please give details of number of plots in the legend.

We have added the number of plots

236-243 – As above, give details of number of plots represented in the figure legends

We have added the number of plots

249-250 – Rather than just significantly different, state which is higher/lower in terms of Lepidoptera species number and Shannon diversity in Jambi versus Kalimantan for clarity.

We have revised the sentence

253 -254 – Table 4 legend – Please give details of the number of plots in the legend, as above for other tables. Write abbreviations like “No. par” in full to avoid confusion.

We have revised the legend and written in full in the table

257 – Table 5 legend – please give details of number of plots in the legend.

We have added the number of plots

270-272 – As above – please say in which direction this marginal effect is (ie it is a negative association).

We have revised the sentence

262-277 – Section “Effect of host abundance, tree age, and understorey vegetation on parasitoid abundance”. As above, I am not totally sure whether it is number of Fornicia species or abundance– I think the former. Please clarify.

We have revised the sentence and included all parasitoid wasps in the analysis

278 – Table 7 legend – as above, please give N values in legend.

We have added the value

Discussion

299 -309 – Suggest adding a line or two to the end of this section, stating that the different results between the two sections of the study could be due to different methods and metrics used, with direct collection and raising of parasitoids likely to be more sensitive to the effects of oil palm age.

We have added a sentence

310-311 – “understorey vegetation, also did not affect the structure of host-parasitoid food webs in oil palm plantations.” Although number of larvae species did decrease with more cover? I think this should be added in here and discussed briefly, as it has interesting management implications.

We have discussed in the paragraph

332-342 – It would be useful to add some brief details of what the two study regions were like in the methods – ie did they differ in amount of forest cover etc, as you suggest could be driving results here?

We have added the different condition of oil palm plantations between Central Kalimantan and Jambi

Conclusion

355-368 – I think the findings also suggest that it might be best to replant oil palm in smaller blocks, creating a more diverse age-structure in oil palms?

We did not analyse about the effect of block size and the diverse age-structure on the host-parasitoid food web. Therefore, we do not suggest in the conclusion.

I hope that these additional comments help with this interesting and important paper.

Thank you very much for your inputs and comments

---

## [Decision Letter · Decision Letter 2]

20 Sep 2023

PONE-D-22-26542R2Long-term changes as oil palm plantation age simplify the structure of host-parasitoid food websPLOS ONE

Dear Dr. Rizali,

Thank you for submitting your manuscript to PLOS ONE. After careful consideration, we feel that it has merit but does not fully meet PLOS ONE’s publication criteria as it currently stands. Therefore, we invite you to submit a revised version of the manuscript that addresses the points raised during the review process.

First of all, I apologize for taking so long to revise your work. Only minor suggestions that had no impact on the final form of the work were made in the final review. However, the reviewer also requested that the text's English quality be improved. I contend strongly that the quality of the English used in writing the manuscripts must be checked and corrected.

We look forward to receiving your revised manuscript.

Kind regards,

Lucas D. B. Faria

Academic Editor

PLOS ONE

Journal Requirements:

Additional Editor Comments:

First of all, I apologize for taking so long to revise your work. Only minor suggestions that had no impact on the final form of the work were made in the final review. However, the reviewer also requested that the text's English quality be improved.

I contend strongly that the quality of the English used in writing the manuscripts must be checked and corrected.

Reviewers' comments:

Reviewer's Responses to Questions

**Comments to the Author**

1. If the authors have adequately addressed your comments raised in a previous round of review and you feel that this manuscript is now acceptable for publication, you may indicate that here to bypass the “Comments to the Author” section, enter your conflict of interest statement in the “Confidential to Editor” section, and submit your "Accept" recommendation.

Reviewer #1: All comments have been addressed

2. Is the manuscript technically sound, and do the data support the conclusions?

Reviewer #1: Yes

3. Has the statistical analysis been performed appropriately and rigorously? 

Reviewer #1: Yes

4. Have the authors made all data underlying the findings in their manuscript fully available?

Reviewer #1: Yes

5. Is the manuscript presented in an intelligible fashion and written in standard English?

Reviewer #1: No

6. Review Comments to the Author

Reviewer #1: I’m sorry to be slow doing this review. The authors have addressed my comments thoroughly and I’m happy for this paper to be accepted. It contributes valuable and detailed information about the ecological impacts of oil palm aging, and has important implications for pest management in these systems. If of help, I have some minor suggestions:

Methods – 147 – “similar heights” it would be good to give heights in text.

Table 1 – I think HL in the table should be HH?

Table 3 – please clarify what LL and HH means in the figure legend for this too, or rephrase to “Abundance of lepidopteran larvae” “Number of species of lepidopteran larvae” etc. for clarity.

7. PLOS authors have the option to publish the peer review history of their article (what does this mean?). If published, this will include your full peer review and any attached files.

Reviewer #1: **Yes: **Edgar Turner

---

## [Author Response · Author response to Decision Letter 2]

24 Sep 2023

Dear Reviewer,

We would like to resubmit our manuscript ID PONE-D-22-26542R2 entitled “Long‐term changes as oil palm plantation age simplify the structure of host‐parasitoid food webs" for publication in PLOS ONE.

We thank you for the constructive comments and we have revised the manuscript accordingly. We also have did proofreading to improve the quality of our manuscript.

This document contains your comments and our response in italics. Where we thought recommended changes were not appropriate, we give the reason in the response.

We hope you will find our manuscript suitable for publication in PLOS ONE.

Sincerely,

Akhmad Rizali

Reviewer #1:

I’m sorry to be slow doing this review. The authors have addressed my comments

thoroughly and I’m happy for this paper to be accepted. It contributes valuable and detailed

information about the ecological impacts of oil palm aging, and has important implications for pest management in these systems. If of help, I have some minor suggestions:

Methods – 147 – “similar heights” it would be good to give heights in text.

We did not measure exactly the heights of oil palm trees on each plot. This statement emphasizes that similar height of oil palm trees within a plot and may different with the height of oil palm trees with other plots

Table 1 – I think HL in the table should be HH?

We have replaced HH to HL in the title of Table 1

Table 3 – please clarify what LL and HH means in the figure legend for this too, or rephrase to

“Abundance of lepidopteran larvae” “Number of species of lepidopteran larvae” etc. for clarity.

We have added the description in the title of Table 3 and also added the abbreviation in the title of Figure 1

---

## [Editor Report · Decision Letter 3]

25 Sep 2023

Long-term changes as oil palm plantation age simplify the structure of host-parasitoid food webs

PONE-D-22-26542R3

Dear Dr. Rizali,

We’re pleased to inform you that your manuscript has been judged scientifically suitable for publication and will be formally accepted for publication once it meets all outstanding technical requirements.

Kind regards,

Lucas D. B. Faria

Academic Editor

PLOS ONE
---

## [Editor Report · Acceptance letter]

2 Oct 2023

PONE-D-22-26542R3 

Long-term changes as oil palm plantation age simplify the structure of host-parasitoid food webs 

Dear Dr. Rizali:

I'm pleased to inform you that your manuscript has been deemed suitable for publication in PLOS ONE. Congratulations! Your manuscript is now with our production department. 

Kind regards, 

on behalf of

Dr. Lucas D. B. Faria 

Academic Editor

PLOS ONE